 # Aletheia: What Makes RLVR For Code Verifiers Tick?

## Abstract

Multi-domain thinking verifiers trained via Reinforcement Learning with Verifiable Rewards (RLVR) are a cornerstone of modern post-training. However, their adoption in code generation has lagged behind that of execution feedback due to the prohibitive costs of the full RLVR pipeline. In this work, we ablate three primary choices along the performance–cost trade-off in RLVR: intermediate thinking traces, learning from negative samples, and on-policy training. We introduce **Aletheia**, a controlled, execution-grounded testbed to facilitate a decontaminated analysis of code verifier training recipes across disparate model sizes and covariate shifts across two common verifier application scenarios. Our analysis reveals that the optimal training recipe is scale-dependent: on-policy learning is the primary performance driver for small verifiers, whereas the thinking budget becomes the most vital factor at larger scales. Negative samples play a key role in stabilizing training at large sizes have a constant impact on top-1 selection, but are increasingly important for ranking performance as size increases. Our Pareto optimality analysis demonstrates that eliminating on-policy training at larger model scales could yield a verifier that performs comparably to the full RLVR recipe. Furthermore, we find that eschewing thinking traces is a compute-efficient strategy at lower budgets, offering a strong trade-off between training cost and verifier accuracy. We validate our findings across a Best-of-N deployment setting and two external reward model benchmarks, demonstrating that our findings generalize beyond the controlled testbed. Ultimately, our work offers empirical guidance toward training cost-efficient code verifiers and takes a step toward their wider adoption in post-training pipelines for code generation.

## 1 Introduction

There has been a strong uptick in the adoption of coding assistants that use agentic harnesses such as Claude Code (Anthropic, 2025). Their daily use by software engineers is driven by recent increases in capability via improved post-training (Yang et al., 2025; Research et al., 2026). Most post-training recipes require that LLM-generated code be verified against runtime signals (Le et al., 2022; Shojaee et al., 2023; Gehring et al., 2025; Liu et al., 2023a). However, self-contained executable codes with accompanying test-cases are a scarce resource, even for curated competitive programming datasets, and manual creation does not scale Wang et al. (2025d). While automatic test-case generation is a common solution (Li et al., 2022; Liu et al., 2023b; Li et al., 2023a), it struggles with test coverage and the inherent difficulty of specifying assertions for open-ended tasks. Alternatives like self-contained environment creation (Jain et al., 2025b; Xie et al., 2024) and world modelling (FAIR et al., 2025) can be challenging for compiled languages without mature package managers.

In this work, we revisit surrogate code-execution verifiers (Ni et al., 2023; Li et al., 2025b; Zeng et al., 2025; Shi et al., 2022; Zhang et al., 2023b): models trained to score code snippets based on execution outcomes without actually executing them. In addition to removing the dependence on high-quality test cases, these verifiers obviate code execution and environment-setup overheads in downstream applications like Reinforcement Learning with Verifiable Rewards (RLVR; Zhu et al., 2026) and Best-of-N (BoN) inference (Zeng et al., 2025). Such verifiers can additionally leverage the code understanding and generalization capabilities of LLMs to provide granular feedback signals for long-horizon tasks where sparse episodic rewards may fail to adequately guide convergence (Cui et al., 2025).

---

We open source our code and datasets at: `https://anonymous.4open.science/r/aletheia-2026/`

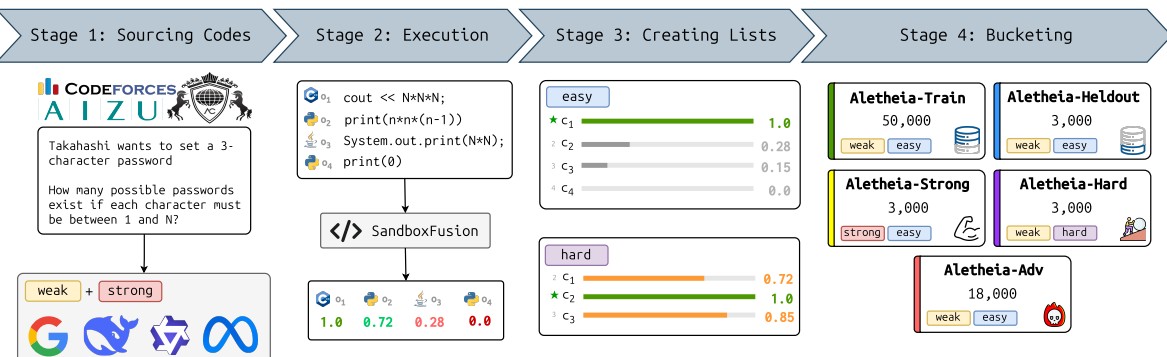

Figure 1: **Aletheia testbed curation pipeline.** We follow a four-stage procedure: (1) generating solutions for competition-level programming questions from `CodeContests⁺` (Wang et al., 2025d) from a pool of **Weak** and **Strong** open-source LLMs; (2) obtaining ground-truth pass rates (`PRs`) for the obtained codes through execution using SandboxFusion; (3) constructing lists of `2-5` candidates where exactly one is fully correct. These lists are either **Easy** (second best $PR < 0.5$) or **Hard** (second best $PR \in [0.7, 0.9]$); and (4) partitioning the resulting data into completely disjoint training and evaluation sets across three covariate shifts: strong generators, hard comparisons, and adversarial prompts. Refer to Section 2.1 for a detailed description.

**The lack of RLVR adoption for code verifiers.** Recently, generative verifiers have proliferated in post-training pipelines for reasoning-heavy domains such as math and science (Liu et al., 2025b; Ma et al., 2025; Cen et al., 2025). Often, these verifiers are trained to reason before answering using RLVR (Chen et al., 2025d; Huang et al., 2025), which boosts interpretability (Gunjal et al., 2025) and mitigates reward hacking (Chen et al., 2025b). However, these advances have made fewer inroads into code verifiers, which are predominantly encoder-only regression models (Zhu et al., 2026; Zhang et al., 2023b; Shi et al., 2022). Although such models can approximate code execution outcomes (Akhauri et al., 2025), they remain brittle against adversarial and semantic-preserving transformations even at scale (Haroon et al., 2025; Lyu et al., 2025), pointing to the need for richer, generative approaches. Yet the lack of adoption is unsurprising: the full RLVR recipe is expensive and demands intricate orchestration of rollout, behavior, and reference policies. Thus, we study verifier training along three axes where RLVR differs from cheaper post-training recipes, namely: generating long intermediate reasoning traces (**Thinking**), learning from both positive and negative samples (**Negatives**), and learning from data generated by an updated policy (**Online**).

Although these RLVR components have been studied independently for generator training (Tajwar et al., 2024; Lanchantin et al., 2025; Zhu et al., 2025), no equivalent analysis exists for verifiers. This gap is significant due to the well-documented inconsistencies between the generative and verification abilities of LLMs (Li et al., 2023b; Rodriguez et al., 2025; Song et al., 2025). Unlike generators, evaluating verifiers requires going beyond downstream task accuracy. Razin et al. (2026) show that even a perfectly accurate verifier can induce a flat loss landscape and stall learning. Moreover, LLM verifiers are known to break down under distribution shifts (Eisenstein et al., 2024) and adversarial prompting (Moon et al., 2025; LAM et al., 2025), which can corrupt the generator being supervised. The optimal code verifier training recipe also likely varies with scale (Kaplan et al., 2020; Hoffmann et al., 2022) and depends on the application, such as BoN selection or use as an RL reward model (Wen et al., 2025; Kim et al., 2025).

**The need for an evaluation testbed for verifiers.** Although integrating verifiers into their downstream use cases seems like a natural way to evaluate verifiers, it is noisy and the resulting policy improvement is a poor proxy for reward quality. This is especially true for verifiers used as RL reward models. Such methods are known to succeed only when the base model learns certain task-specific primitives during earlier training phases (Setlur et al., 2026; Zhang et al., 2025a; Wang et al., 2025c), making downstream performance an amalgam of prior capability, reward signal, and the optimization process (Wen et al., 2025; Kim et al., 2025). These confounds are so stark that certain models improve even under random rewards (Shao et al., 2025).

Additionally, the pipeline is expensive and slow to iterate on, which significantly hurts reproducibility and slows progress (Frick et al., 2025). Evidence suggests that the early steps in RLVR merely improve sampling efficiency (Yue et al., 2025; Wu et al., 2025) and true expansion of reasoning capabilities requires prolonged

training (Liu et al., 2025a; Yao et al., 2025b), which puts a principled evaluation out of reach for most reasonable budgets. Even recent RL scaling studies require discarding the first ∼1.5k GPU-hours of every run (Devvrit et al., 2026), much like pre-training scaling studies (Li et al., 2025a; Porian et al., 2024).

To sidestep these limitations, we create **Aletheia**, an execution-grounded testbed for evaluating verifier training recipes. We mirror downstream evaluation scenarios through a decontaminated testbed that enforces a strict training – evaluation partition, isolating algorithmic robustness to out-of-distribution (OOD) scenarios from simple data exposure. We ablate the three RLVR components mentioned earlier across three covariate shifts commonly encountered in downstream evaluations: stronger generators than the ones seen during training (`Aletheia-Strong`), incorrect solutions that are semantically very close to the correct one (`Aletheia-Hard`), and adversarially modified code snippets (`Aletheia-Adv`). We validate our findings across three model sizes (`1.5/7/14B`), uncovering scale-dependent training dynamics. We construct our testbed in accordance with prior work on reward model evaluations (Feng et al., 2025; Wen et al., 2025; Kim et al., 2025) and evaluate each verifier's ability to select the best candidate (top-1 selection) and reproduce the full ranked order (reranking) as a proxy for BoN and RL performance respectively. These metrics are evaluated over lists of 2 – 5 codes, and better correlated with downstream performance than paired accuracy (Wen et al., 2025).

Our analysis reveals that while RLVR is the best-performing method to train verifiers in most evaluation settings, the contribution of each ablated component varies with scale. Across both downstream application scenarios, we find that on-policy learning is critical for small verifiers, but its contribution diminishes as model size increases. Conversely, thinking traces offer limited benefits at smaller scales but become essential for `14B` models. Meanwhile, negative samples provide a near-consistent boost to all sizes for top-1 selection but a monotonic boost for reranking, and play a critical role in stabilizing training at larger scales. Utilizing additional compute for verifiers at inference time using self-consistency yields modest gains and cannot compensate for any of the core components we analyze.

Additionally, we study the Pareto optimality of each ablated recipe with respect to cost and performance. `DPO-Think-14B` is competitive with the full `GRPO-Think` recipe at `5.2×` lower cost, and is the only 14B method on the Pareto frontier across all evaluations simultaneously. The full GRPO recipe is warranted only when peak performance across generator shifts and adversarial perturbations is required. For `Aletheia-Hard`, practitioners should favor `DPO-Think-14B` for top-1 selection or `RAFT-14B` for reranking, as on-policy training provides no cost-adjusted benefit. `GRPO-Instruct-7B` is a good low-budget option on the Pareto frontier for all evaluations, but has a weaker absolute performance than `DPO-Think-14B`. We summarize our contributions as follows.

- We introduce **Aletheia**, an execution-grounded testbed that enables a decontaminated evaluation across three covariate shifts: stronger generators, harder comparisons, and adversarial responses (Section 2).
- We ablate the contributions of three core RLVR components to code verifier performance across different model sizes (`1.5/7/14B`): thinking traces, on-policy learning, and negative samples (Section 3).
- We conduct a cost-benefit analysis of the RLVR training recipe for robust code verifiers (Section 4).
- We validate our findings across a downstream Best-of-N selection and two external benchmarks (Section 5).

## 2 Experimental Setup

### 2.1 Testbed Creation

A code verifier's value is ultimately determined by its downstream use, either in BoN inference or as a reward model inside an RLVR pipeline. Evaluating verifiers against either of these uses, however, is far from straightforward. The de-facto offline metric, paired accuracy on RewardBench-style benchmarks (Lambert et al., 2025; Liu et al., 2024b; Tan et al., 2024), is cheap but only weakly predictive of downstream performance: it is insensitive to score separation across the full candidate list (Razin et al., 2026), to discrimination under near-correct distractors (Kim et al., 2025; Feng et al., 2025), and to the distributional coverage required as the policy drifts from the verifier's training distribution (Eisenstein et al., 2024; Wen et al., 2025).

To reliably evaluate verifier training recipes, we construct **Aletheia**: a controlled, decontaminated, execution-grounded testbed. Each prompt is a listwise selection problem, which disincentivizes models from gaming rewards via random guessing and correlates better with downstream performance than paired accuracy (Wen et al., 2025). Following prior work on reliable verifier evaluation (Kim et al., 2025), we source codes from

Table 1: **Descriptive statistics for our datasets.** Code lengths are measured in tokens and averaged over the lists within a single instance. Average similarities are computed as the cosine similarities between the embeddings of the codes in each list.

| Name | #Rows | #Qtns | Code Len. | Code Sim. |
|------|-------|-------|-----------|-----------|
| Aletheia-Train | 50,000 | 1247 | 208 | 0.896 |
| Aletheia-Heldout | 3,000 | 456 | 236 | 0.893 |
| Aletheia-Strong | 3,000 | 151 | 287 | 0.898 |
| Aletheia-Hard | 3,000 | 137 | 186 | 0.931 |
| Aletheia-Adv | 18,000 | 456 | 246 | 0.882 |

Table 2: **Generators used for our datasets.** $\Delta$Score is the difference in BigCodeBench-Instruct scores (Zhuo et al., 2025) between the **Strong** and **Weak** models, representing the gap in their abilities. We use the -Instruct variants for all the generators.

| Model Family | Weak | Strong | $\Delta$Score |
|--------------|------|--------|---------------|
| deepseek-ai/deepseek-coder | 6.7B | 33B | 6.5 |
| google/gemma2 | 9B | 27B | 5.1 |
| meta/llama-3.1 | 8B | 70B | 13.3 |
| qwen/qwen2.5-coder | 7B | 32B | 8.6 |

a wide range of model families and sizes (Table 2), but maintain a single generator within a single prompt to mimic downstream evaluation (Feng et al., 2025). Concretely, given a coding problem $\mathbf{x}$ and $N$ distinct candidate solutions $\mathbf{C} = \{\mathbf{c}_n\}_{n=1}^N$ with execution-derived pass rates $\mathbf{P} = \{\mathrm{p}_n\}_{n=1}^N$, the verifier generates $\mathbf{y} = (\mathbf{z}, \mathbf{o})$: a reasoning trace $\mathbf{z}$ followed by a predicted index $\mathbf{o} \in \{1, \dots, N\}$ identifying the best candidate. By construction, exactly one candidate is fully correct ($\max_n \mathrm{p}_n = 1$), so the ground truth is unambiguous and the reward $\mathbf{1}[\mathbf{o} = \arg\max_n \mathrm{p}_n]$ is trivially computable.

We source competitive programming problems from CodeContests$^+$ (Wang et al., 2025d), each with $\approx 25$ synthetic test cases. We only include high-quality problems with true positive rates and true negative rates > 0.9 against a pre-evaluated user-submitted solution pool, and discard samples with < 5 test cases or a time limit > 3 seconds. This gives us 4903 programming questions. We generate 50 completions for each problem in Python, C++, and Java at a sampling temperature of 1.0 using a mix of **Weak** and **Strong** LLMs (Table 2). The resulting code snippets are executed using SandboxFusion[1] to calculate their pass rates (PRs); the percentage of test cases passed for each code. We construct lists of 2 − 5 solution codes with distinct PRs. We divide these lists into **Easy** and **Hard** buckets based on their PRs; **Easy** contains lists where $\mathrm{PR}_{\mathrm{incorrect}} \in [0, 0.5]$ and **Hard** has $\mathrm{PR}_{\mathrm{incorrect}} \in [0.7, 0.9]$, making them harder to distinguish from the correct code.

We subsample the **Weak**-**Easy** bucket to 50,000 instances, dubbed **Aletheia-Train**. Additionally, we create four evaluation datasets, evenly distributed by list length and programming language:

- **Aletheia-Heldout.** An in-distribution evaluation set containing **Easy** comparisons by **Weak** models.
- **Aletheia-Strong.** **Easy** comparisons by **Strong** models, which tests the verifier's robustness to a shift in the generator's capability, without altering the quality of the codes being compared (Zhou et al., 2025).
- **Aletheia-Hard.** **Hard** comparisons generated by **Weak** models. Verifier performance on this dataset indicates their ability to generalize from easy to hard (Hase et al., 2024; Sun et al., 2024).
- **Aletheia-Adv.** It evaluates the adversarial robustness of our verifiers. We apply three positive and negative modifications to the incorrect and correct codes in Aletheia-Heldout respectively (see Section C), based on prior work on biases in LLM judges (LAM et al., 2025; Hwang et al., 2025; Moon et al., 2025).

All OOD evaluations introduce a unidirectional shift from the training data. We further reduce contamination by ensuring no overlap between training and evaluation coding problems. The prompts used in this work are listed in Section F. We summarize dataset statistics in Table 1. We calculate the average code similarities in our datasets by encoding them using Qwen3-Embedding-8B (Zhang et al., 2025b), which achieves stellar performance on text embedding benchmarks like MTEB (Muennighoff et al., 2023; Enevoldsen et al., 2025). The codes within a single prompt are quite similar by virtue of being generated from the same model, with all similarity scores being > 0.88. The codes in Aletheia-Hard particularly stand out, with an average similarity of 0.93 and a smaller average length of 186 tokens. Since all codes in this dataset pass at least 70% of the test cases, they are semantically very close to the correct solution and thus harder to distinguish.

## 2.2 Evaluation Metrics

The Aletheia testbed is carefully designed to evaluate reward models in their two most common downstream usecases via complementary metrics. Crucially, our task formulation matches BoN inference almost exactly:

---

[1] bytedance/SandboxFusion

selecting the $PR=1.0$ winner from a generator-matched pool across a variable number of candidates. Moreover, recent work suggests that accuracy evaluated over multiple comparisons and across a range of response quality predicts BoN performance better than paired accuracy (Wen et al., 2025). Thus, we report the average top-1 selection accuracy (`ListAcc`) across all evaluation datasets as a predictor of downstream BoN performance. During evaluation, we generate responses using a temperature of `0.6` and `top-p = 0.95` nucleus sampling. We draw 16 generations per prompt and report 95% confidence intervals over them in all tables.

However, accuracy alone is often insufficient to predict the utility of a verifier as an RL reward-model (Wen et al., 2025; Feng et al., 2025). As mentioned in Section 1, direct RL integration may also not be indicative of reward quality and is prohibitively expensive. Thus, we evaluate our verifiers' ability to reconstruct the full ranked order of $N$ candidates rather than selecting the best candidate, which may better predict downstream RL performance (Wen et al., 2025). We adopt Kendall's $\tau$-b (`Kτ`) as our reranking metric. For each list of $N$ candidates, we issue all $\binom{N}{2}$ pairwise comparisons as independent verifier generations and parse each verdict to identify the winning candidate. We score every candidate $\mathbf{c}_n$ by its number of pairwise wins and compute

$$\mathsf{K}\tau = \frac{n_C - n_D}{\sqrt{(n_C + n_D + T_w)(n_C + n_D + T_p)}},$$

where $n_C$ and $n_D$ are the numbers of candidate pairs ranked concordantly and discordantly by the predicted win counts against the execution-derived pass rates $\mathbf{P}$, and $T_w$, $T_p$ count pairs tied in the predicted win counts and in $\mathbf{P}$ respectively ($T_p = 0$ in our case). We use the tie-corrected $\tau$-b rather than $\tau$-a because candidates routinely tie in predicted win counts. We report `Kτ` averaged over lists in each evaluation set. A `Kτ = +1` implies a perfectly ordered list, `0` an uninformative ordering, and `−1` indicates a fully reversed one. Unlike top-1 selection accuracy, `Kτ` uses every pair in the candidate list and remains discriminative even when correct and incorrect codes have near-identical pass rates, which captures the score-separation property required by downstream policy-gradient training (Razin et al., 2026).

### 2.3 Training Details

We validate our findings across a wide range of model sizes and training parameter counts, including `1.5B`, `7B`, and `14B` for each method. Unless explicitly mentioned, we initialize each method from the `DeepSeek-R1-Distill-Qwen2.5` models (DeepSeek-AI, 2025) because they have been warm-started to generate reasoning traces before answering. To ensure a fair comparison, all methods are trained for an identical number of gradient updates. For on-policy methods, we generate `16` responses at a high sampling temperature of `1.0` and award a `+1` to generations that identify the correct candidate, and `0` otherwise. We also apply a `−1` penalty to generations that violate the format. We provide a detailed description of our training setup in Section A and experiment with alternate reward formulations in Section B.

## 3 Research Questions and Results

Our analysis employs a series of controlled experiments to analyze the contributions of Thinking, Negatives, and Online components. We use `GRPO` (Shao et al., 2024) as a baseline that includes all three components and compare it with algorithms that lack the single component under study. This design choice minimizes confounding factors to isolate each component's individual contribution as much as possible. We study the effect of removing two components in Section E.1.

### 3.1 RQ1: Do Code Verifiers Need to Generate Long Reasoning Traces?

> **Summary Of Findings For Best-of-N 1**
> - The contribution of thinking traces to verifier performance increases monotonically with model scale.
> - Training reasoning budget follows a similar pattern: 1.5B saturates beyond 8k, but 7-14B continue improving up to 16k.
> - Thinking traces are crucial for **Easy**-to-**Hard** generalization.
> - Reasoning-style traces enable models to utilize increased inference compute.

> **Summary Of Findings For RL 1**
> - The contribution of thinking traces to RL reward model performance similarly increases with model size.
> - Training reasoning budget yields continued gains in `Kτ` up to 16k for all verifier sizes.
> - `Kτ` and `ListAcc` agree on general trends, but the magnitude of each effect varies.

Table 3: **Results for ablating thinking trace generation (💡).** We report list accuracy and Kendall $\tau$ scores for BoN and RL, respectively. For both metrics, thinking-style traces offer little benefit to small models but are essential for larger models. Increasing the reasoning budget to 16k is almost always useful, but the style of traces alone is most impactful for larger models. Thinking is especially critical for `Aletheia-Hard`. Subscripts are 95% confidence intervals.

| | | | | 💡 Thinking | | 🐦 Negatives | ⊘ Online | | | | |
|---|---|---|---|---|---|---|---|---|---|---|---|
| | | | Aletheia-Heldout | | Aletheia-Strong | | Aletheia-Hard | | Aletheia-Adv | | Average | |
| | Method | Size | Cost ($) | BoN | RL | BoN | RL | BoN | RL | BoN | RL | BoN | RL |
| – | Random | | – | 32.08 | 0.00 | 32.08 | 0.00 | 32.08 | 0.00 | 32.08 | 0.00 | 32.08 | 0.00 |
| | GRPO-Instruct | 1.5B | 0.587 | $38.78_{0.85}$ | $11.90_{2.18}$ | $40.51_{0.85}$ | $13.09_{2.29}$ | $31.22_{0.81}$ | $4.11_{2.37}$ | $30.99_{0.33}$ | $2.16_{2.24}$ | $35.41_{0.71}$ | $7.82_{2.27}$ |
| | GRPO-Think-4k | | 1.208 | $42.73_{0.64}$ | $18.48_{2.38}$ | $40.70_{0.61}$ | $13.08_{2.38}$ | $36.32_{0.57}$ | $5.36_{2.40}$ | $31.15_{0.25}$ | $2.32_{2.44}$ | $37.76_{0.52}$ | $9.81_{2.40}$ |
| | GRPO-Think-8k | | 2.491 | $46.82_{0.62}$ | $22.90_{2.39}$ | $43.65_{0.59}$ | $17.56_{2.40}$ | $41.61_{0.61}$ | $10.41_{2.41}$ | $38.09_{0.25}$ | $9.50_{2.54}$ | $42.55_{0.52}$ | $15.09_{2.43}$ |
| | GRPO-Think-16k | | 7.806 | $49.58_{0.65}$ | $27.49_{2.32}$ | $46.09_{0.63}$ | $23.57_{2.33}$ | $40.74_{0.64}$ | $8.92_{2.48}$ | $40.97_{0.26}$ | $16.45_{2.53}$ | $44.38_{0.55}$ | $19.11_{2.42}$ |
| | GRPO-Instruct | 7B | 2.069 | $57.74_{0.88}$ | $35.42_{2.00}$ | $51.80_{0.89}$ | $29.08_{2.35}$ | $38.59_{0.87}$ | $9.88_{2.40}$ | $52.20_{0.37}$ | $29.20_{2.18}$ | $50.07_{0.75}$ | $25.90_{2.23}$ |
| | GRPO-Think-4k | | 3.561 | $59.54_{0.71}$ | $37.40_{2.25}$ | $55.00_{0.70}$ | $32.03_{2.34}$ | $46.73_{0.74}$ | $14.92_{2.40}$ | $44.04_{0.30}$ | $21.84_{2.50}$ | $51.32_{0.61}$ | $26.55_{2.37}$ |
| | GRPO-Think-8k | | 7.179 | $65.03_{0.57}$ | $35.22_{2.09}$ | $56.96_{0.57}$ | $32.40_{2.41}$ | $53.16_{0.65}$ | $17.81_{2.40}$ | $52.03_{0.26}$ | $28.15_{2.35}$ | $56.76_{0.51}$ | $28.40_{2.31}$ |
| | GRPO-Think-16k | | 15.101 | $74.81_{0.57}$ | $51.52_{1.73}$ | $67.28_{0.60}$ | $46.30_{2.03}$ | $53.11_{0.69}$ | $20.31_{2.44}$ | $65.04_{0.26}$ | $46.16_{2.14}$ | $65.05_{0.53}$ | $41.07_{2.08}$ |
| | GRPO-Instruct | 14B | 9.463 | $63.45_{0.82}$ | $41.70_{1.98}$ | $55.11_{0.84}$ | $31.01_{2.17}$ | $44.15_{0.84}$ | $16.97_{2.32}$ | $54.24_{0.35}$ | $27.51_{2.19}$ | $54.26_{0.71}$ | $29.30_{2.17}$ |
| | GRPO-Think-4k | | 8.558 | $73.23_{0.67}$ | $46.14_{1.92}$ | $64.95_{0.70}$ | $39.74_{2.01}$ | $54.56_{0.77}$ | $21.34_{2.31}$ | $58.09_{0.31}$ | $31.46_{2.27}$ | $62.69_{0.61}$ | $34.67_{2.13}$ |
| | GRPO-Think-8k | | 14.900 | $78.37_{0.60}$ | $50.23_{1.86}$ | $69.87_{0.65}$ | $41.73_{1.98}$ | $61.74_{0.73}$ | $27.45_{2.30}$ | $65.71_{0.29}$ | $39.39_{2.12}$ | $68.91_{0.57}$ | $39.70_{2.06}$ |
| | GRPO-Think-16k | | 36.992 | $88.02_{0.45}$ | $60.10_{1.64}$ | $83.65_{0.49}$ | $60.21_{1.74}$ | $66.84_{0.70}$ | $30.95_{2.26}$ | $83.67_{0.21}$ | $56.84_{1.63}$ | $80.54_{0.46}$ | $52.03_{1.82}$ |

**Background.** Thinking traces significantly boost LLM performance (Wei et al., 2022; Kojima et al., 2022), but the source of these gains is ambiguous: several works find no causal relation between the model's CoT and final answer (Turpin et al., 2023; Aljohani et al., 2025), casting doubt on the notion that the generated tokens allow the model to *think* before answering. This behavior is less common, but still prominent in Large Reasoning Models (LRMs) (Chua & Evans, 2025). Thus, long intermediate chains may not directly influence response quality (Stechly et al., 2025; Kambhampati et al., 2025), sparking interest in generating shorter intermediate tokens (Arora & Zanette, 2025; Sui et al., 2025). We quantify the impact of deeper thinking on verifier quality through a controlled ablation study.

**Setup.** We evaluate the impact of generating thinking traces on verifier quality by comparing short chain-of-thought (CoT) with longer reasoning-style traces. We also study the impact of varying $B_{tr} = \text{len}(\mathbf{z})$: the maximum permitted length of reasoning traces during training. We train four models: `GRPO-Instruct` with $B_{tr} = 4096$, and three `GRPO-Think` variants with $B_{tr} \in \{4096, 8192, 16384\}$. `GRPO-Instruct` is initialized from the `Qwen2.5-Instruct`, which does not generate thinking traces by default. We report BoN and RL performance trends using the metrics discussed in Section 2.2, along with the average costs per step, assuming \$10.6 per H200-hour[2]. We also study the effects of self-consistency (Wang et al., 2023) on BoN performance.

**Findings as a Best-of-N selector.** `GRPO-Think-16k` outperforms all other verifiers in Table 3, but the gap to `GRPO-Instruct` varies with scale. `GRPO-Instruct` and `GRPO-Think-4k` differ by $\leq 2.4$ BoN points on average for `1.5-7B` verifiers, indicating the *style* of intermediate trace makes little difference at smaller scale. At `14B`, the same comparison goes to `8.4` points. Expanding the training reasoning budget follows the same pattern: for `1.5B`, the `4k`→`8k` doubling adds `4.8` BoN points, but `8k`→`16k` adds only `1.8`, indicating diminishing returns. In contrast, `7-14B` models keep climbing up to `16k`, with gains of `8.3` and `11.6` BoN points respectively at the `8k`→`16k` step. These trends are likely driven by larger models being generally better at utilizing thinking primitives (Gandhi et al., 2025) and longer contexts (Hsieh et al., 2024; Liu et al., 2024a).

Across sizes, the verifiers stay robust to shifts in generator capabilities on `Aletheia-Strong`, losing only $\approx 5.2$ BoN points across all models. This reproduces the scalable-supervision pattern of Burns et al. (2024) and contrasts with earlier reports on verification tasks (Zhou et al., 2025). Similar to the average trend, the style of intermediate traces has minimal influence on `1.5-7B` models, and increasing $B_{tr}$ to `8k` and `16k` tracks the same trend for shifts in generator capability.

---

[2] 🌐 jarvislabs.ai/h200-price

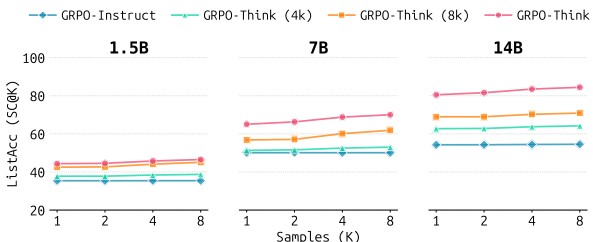
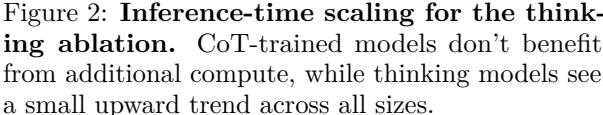

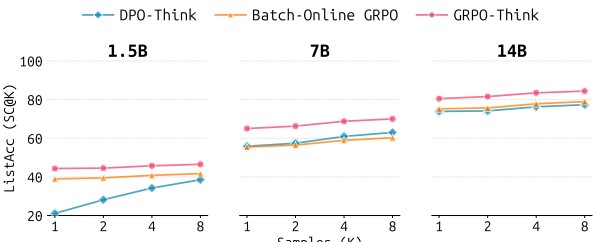

Figure 2: **Inference-time scaling for the thinking ablation.** CoT-trained models don't benefit from additional compute, while thinking models see a small upward trend across all sizes.

Figure 3: **Inference-time scaling for the online-offline ablation.** The offline – online gap narrows with increasing K, but does not close. `DPO-Think` can surpass `BO-GRPO` at larger inference budgets.

Overall, the `1.5B` models are sensitive to adversarial perturbations, performing worse than the random baseline for $B_{tr}$=4k. This drop in performance is mitigated by scaling the reasoning budget to `16k`, increasing the model size, or both. Crucially, simply switching the *style* of thinking traces has only a small impact on performance, and can even hurt the `7B` model. We observe robustness to adversarial perturbations only at the larger reasoning budgets and model sizes, contradicting the general notion that reasoning traces make models less vulnerable to judging biases (Wang et al., 2025a). The training – evaluation partition in `Aletheia` allows us to attribute these trends directly to the training recipe, without contamination confounds.

We observe consistently large performance drops on `Aletheia-Hard`, suggesting that easy-to-hard generalization performance is challenging for all model sizes. Contrary to the general trend, simply switching from CoT to reasoning traces improves BoN performance across the board, with additional scaling bringing linear gains. Since the codes in `Aletheia-Hard` are highly similar (c.f. Table 1), verifiers benefit from utilizing reasoning primitives like backtracking and subgoal-setting (Gandhi et al., 2025).

Thinking is also vital for utilizing additional compute at test-time with self-consistency, with CoT traces in `GRPO-Instruct` yielding a flat `SC@K` curve (Figure 2). All `GRPO-Think` variants see a slight upward trend, with the effect being most pronounced at $B_{tr}$=16k. The curves are strictly ordered across all sizes, suggesting that while self-consistency provides modest gains at inference-time, it cannot replace $B_{tr}$ scaling.

**Findings as an RL reward.** The comparison under $K\tau$ paints a very similar picture to the one under `ListAcc`, with `GRPO-Think-16k` emerging as the best verifier for RL training with $K\tau$=52.03 at the `14B` scale. The training budget sweep also tells the same story under $K\tau$ with the algorithm that produces the best selector also generally producing the best ranker, albeit with different magnitudes. Similarly, the trends established for the OOD evaluation datasets on BoN also generally hold for the RL setting. However, there are some notable exceptions to this trend. Although at `1.5B`, increasing $B_{tr}$ from `8k`→`16k` yields diminishing returns for BoN as established earlier, it yields continued gains for RL. On `Aletheia-Adv`, this scaling yields a modest 2.88 BoN points, but a more substantial 6.95 $K\tau$ points. Surprisingly, while increasing $B_{tr}$ from `4k`→`8k` for the `7B` model brings a 5.49 BoN point gain on `Aletheia-Heldout`, it decreases $K\tau$ by 2.18.

### 3.2 RQ2: Is On-policy Learning Essential for Verifier Training?

**Summary Of Findings For Best-of-N 2**
- Off-policy training collapses below random at `1.5B`, but is competitive with semi-online methods at larger sizes.
- The online-offline gap between DPO–GRPO narrows with scale but never closes: from 23.27 BoN points at `1.5B` to 6.65 at `14B`.
- `BO-GRPO` recovers neither `GRPO-Think`'s BoN performance nor offers a meaningful cost saving.
- Inference-time scaling narrows the offline–online BoN gap, most visibly at `1.5B`, but cannot close it.

**Summary Of Findings For RL 2**
- `DPO-Think-1.5B`'s high $K\tau$ scores are an artifact of noisy evaluation due to low response parseability.
- The necessity of on-policy learning for RL training similarly diminishes with scale.
- On-policy training has little effect on easy-to-hard generalization.
- `BO-GRPO` offers no consistent $K\tau$ gain over offline DPO at 7–14B despite higher compute cost.

Table 4: **Results for ablating on-policy learning (⟳).** We report list accuracy and Kendall $\tau$ scores for BoN and RL, respectively. Online learning is important for small verifiers, but its importance diminishes as scale increases. Batch-online methods are similarly useful at small scales but don't help larger models. On-policy is especially irrelevant for **Easy**-to-**Hard** generalization. Subscripts are 95% confidence intervals.

| | | | | 💡 Thinking    ⚒ Negatives    ⟳ Online    ◉ Batch-online | | | | | | | | |
|---|---|---|---|---|---|---|---|---|---|---|---|---|---|
| | | | | Aletheia-Heldout | | Aletheia-Strong | | Aletheia-Hard | | Aletheia-Adv | | Average | |
| | Method | Size | Cost (\$) | BoN | RL | BoN | RL | BoN | RL | BoN | RL | BoN | RL |
| – | Random | | – | 32.08 | 0.00 | 32.08 | 0.00 | 32.08 | 0.00 | 32.08 | 0.00 | 32.08 | 0.00 |
| 💡⚒⟳ | DPO-Think | | 5.951* | $21.70_{0.39}$ | $9.46_{2.63}$ | $19.90_{0.35}$ | $16.00_{2.61}$ | $23.41_{0.37}$ | $12.08_{2.63}$ | $19.37_{0.15}$ | $16.39_{2.58}$ | $21.11_{0.32}$ | $13.48_{2.61}$ |
| 💡⚒◉ | BO-GRPO | 1.5B | 4.322 | $43.13_{0.55}$ | $17.66_{2.48}$ | $39.89_{0.52}$ | $12.53_{2.45}$ | $39.26_{0.54}$ | $7.19_{2.49}$ | $33.83_{0.22}$ | $6.48_{2.54}$ | $38.99_{0.46}$ | $10.97_{2.49}$ |
| 💡⚒⟳ | GRPO-Think | | 7.806 | $49.58_{0.65}$ | $27.49_{2.32}$ | $46.09_{0.63}$ | $23.57_{2.33}$ | $40.74_{0.64}$ | $8.92_{2.48}$ | $40.97_{0.26}$ | $16.45_{2.53}$ | $44.38_{0.55}$ | $19.11_{2.42}$ |
| 💡⚒⟳ | DPO-Think | | 6.403* | $63.75_{0.57}$ | $37.58_{2.05}$ | $55.54_{0.57}$ | $31.89_{2.37}$ | $51.20_{0.62}$ | $16.56_{2.40}$ | $52.94_{0.25}$ | $28.81_{2.38}$ | $55.88_{0.50}$ | $28.71_{2.30}$ |
| 💡⚒◉ | BO-GRPO | 7B | 9.588 | $64.71_{0.60}$ | $41.27_{2.08}$ | $56.18_{0.61}$ | $30.72_{2.31}$ | $52.08_{0.66}$ | $15.68_{2.39}$ | $49.35_{0.27}$ | $26.05_{2.44}$ | $55.46_{0.54}$ | $28.43_{2.30}$ |
| 💡⚒⟳ | GRPO-Think | | 15.101 | $74.81_{0.57}$ | $51.52_{1.73}$ | $67.28_{0.60}$ | $46.30_{2.03}$ | $53.11_{0.69}$ | $20.31_{2.44}$ | $65.04_{0.26}$ | $46.16_{2.14}$ | $65.05_{0.53}$ | $41.07_{2.08}$ |
| 💡⚒⟳ | DPO-Think | | 7.087* | $82.56_{0.52}$ | $51.12_{1.80}$ | $74.39_{0.58}$ | $43.46_{1.85}$ | $67.58_{0.68}$ | $29.90_{2.27}$ | $71.06_{0.26}$ | $44.26_{2.10}$ | $73.89_{0.51}$ | $42.19_{2.00}$ |
| 💡⚒◉ | BO-GRPO | 14B | 31.144 | $83.82_{0.50}$ | $49.42_{1.75}$ | $76.33_{0.56}$ | $41.31_{1.91}$ | $67.34_{0.68}$ | $32.21_{2.22}$ | $73.45_{0.25}$ | $42.01_{2.06}$ | $75.29_{0.50}$ | $41.24_{1.98}$ |
| 💡⚒⟳ | GRPO-Think | | 36.992 | $88.02_{0.45}$ | $60.10_{1.64}$ | $83.65_{0.49}$ | $60.21_{1.74}$ | $66.84_{0.70}$ | $30.95_{2.26}$ | $83.67_{0.21}$ | $56.84_{1.63}$ | $80.54_{0.46}$ | $52.03_{1.82}$ |

**Background.** On-policy learning is perhaps the most widely studied and the most expensive aspect of RLVR training. Despite its effectiveness, on-policy training is very inefficient and often impractical. Thus, practitioners usually resort to introducing some amount of off-policyness to increase training efficiency (Noukhovitch et al., 2025; Piché et al., 2025). There is no consensus on its necessity: some works find it vital to success in RL algorithms (Noukhovitch et al., 2025; Tang et al., 2024; Yu et al., 2025a), while others claim that introducing a certain amount of off-policyness can match or even outperform fully on-policy methods on mathematical reasoning tasks (Lanchantin et al., 2025; Chen et al., 2025a; Song et al., 2024).

**Setup.** We study the impact of this decision through three representative algorithms. `DPO-Think` serves as our purely offline algorithm, and `Batch-online (BO-) GRPO` represents the middle ground between online and offline methods, sampling a batch of responses and performing multiple gradient updates on mini-batches of generated data (Zheng et al., 2025). We present the results in Table 4. For `DPO-Think`, we include the costs of creating an offline preference dataset as detailed in Section A.

**Findings as a Best-of-N selector.** `GRPO-Think` continues to dominate the ablated variants at all model sizes. The offline-online gap narrows with scale, with the `DPO-GRPO` gap shrinking from 23.27 BoN points at 1.5B to just 6.65 at 14B. `DPO-Think` collapses at smaller sizes, performing worse than the random baseline due to degeneration and unparseable verdicts. Specifically, `DPO-Think-1.5B` produces a parseable final answer in only 43% of cases on average (see Table 14 for more details). `BO-GRPO` is a good tradeoff at smaller scales, with lower costs than a fully offline approach, and performance within 6 points of the full `GRPO-Think` recipe.

However, the story flips at 7–14B sizes. `DPO-Think` closes the gap with `BO-GRPO` on BoN while also decreasing costs, and is 4.4× cheaper at 14B. `BO-GRPO` fails to recover `GRPO-Think`'s BoN, and its lower cost is not worth the performance drop, since cheaper alternatives exist. Our observation contradicts that of Lanchantin et al. (2025), who report batch-online methods that match or outperform fully online training on math tasks. We attribute the discrepancy to their use of `Llama-3.1-8B-Instruct` (no thinking traces) and the original GRPO recipe (Shao et al., 2024), while we incorporate improvements detailed in Section A.

The `Aletheia-Strong` and `Aletheia-Adv` evaluations track the general trend almost exactly, except for `DPO-Think-7B` being slightly more robust to adversarial prompts than `BO-GRPO`. On-policy training has no perceptible impact on easy-to-hard generalization at 7–14B, suggesting that a well-curated offline preference dataset may be sufficient for such generalization. Although `DPO-Think-1.5B` gains 1.71 BoN points on `Aletheia-Hard`, it is still worse than random chance and we thus do not consider it a meaningful gain.

Scaling inference-time compute benefits all methods and model sizes (Figure 3). Such scaling particularly benefits `DPO-Think-1.5B`, which almost matches `BO-GRPO`'s BoN at K=8. This behavior is largely driven by a higher parse rate due to increased sampling at inference time. At 7B, additional inference-time compute even

---

*DPO includes the cost for creating the offline dataset.

Table 5: **Results for ablating learning from negatives (🎯).** We report list accuracy and Kendall $\tau$ scores for BoN and RL respectively. Negatives are consistently beneficial across all model sizes for BoN, but more important for large models during RL. **Easy**-to-**Hard** generalization is largely unaffected by negatives. Subscripts are 95% confidence intervals.

| | Method | Size | Cost ($) | Aletheia-Heldout | | Aletheia-Strong | | Aletheia-Hard | | Aletheia-Adv | | Average | |
|---|---|---|---|---|---|---|---|---|---|---|---|---|---|
| | | | | BoN | RL | BoN | RL | BoN | RL | BoN | RL | BoN | RL |
| – | Random | | – | 32.08 | 0.00 | 32.08 | 0.00 | 32.08 | 0.00 | 32.08 | 0.00 | 32.08 | 0.00 |
| 💡🎯⏱ | RAFT | 1.5B | 4.167 | $34.76_{0.48}$ | $14.20_{2.38}$ | $31.88_{0.44}$ | $12.31_{2.48}$ | $33.67_{0.46}$ | $9.06_{2.43}$ | $29.12_{0.19}$ | $12.25_{2.58}$ | $32.30_{0.39}$ | $11.96_{2.47}$ |
| 💡🎯⏱ | GRPO-Think | | 7.806 | $49.58_{0.65}$ | $27.49_{2.32}$ | $46.09_{0.63}$ | $23.57_{2.33}$ | $40.74_{0.64}$ | $8.92_{2.48}$ | $40.97_{0.26}$ | $16.45_{2.53}$ | $44.38_{0.55}$ | $19.11_{2.42}$ |
| 💡🎯⏱ | RAFT | 7B | 6.948 | $60.86_{0.59}$ | $38.82_{2.11}$ | $52.00_{0.58}$ | $31.89_{2.33}$ | $48.84_{0.63}$ | $17.33_{2.41}$ | $49.24_{0.26}$ | $29.36_{2.48}$ | $52.72_{0.52}$ | $29.35_{2.33}$ |
| 💡🎯⏱ | GRPO-Think | | 15.101 | $74.81_{0.57}$ | $51.52_{1.73}$ | $67.28_{0.60}$ | $46.30_{2.03}$ | $53.11_{0.69}$ | $20.31_{2.44}$ | $65.04_{0.26}$ | $46.16_{2.14}$ | $65.05_{0.53}$ | $41.07_{2.08}$ |
| 💡🎯⏱ | RAFT | 14B | 12.906 | $75.55_{0.56}$ | $49.42_{1.82}$ | $66.02_{0.61}$ | $41.31_{2.07}$ | $65.23_{0.65}$ | $32.21_{2.26}$ | $62.03_{0.27}$ | $42.01_{2.11}$ | $67.20_{0.52}$ | $41.24_{2.07}$ |
| 💡🎯⏱ | GRPO-Think | | 36.992 | $88.02_{0.45}$ | $60.10_{1.64}$ | $83.65_{0.49}$ | $60.21_{1.74}$ | $66.84_{0.70}$ | $30.95_{2.26}$ | $83.67_{0.21}$ | $56.84_{1.63}$ | $80.54_{0.46}$ | $52.03_{1.82}$ |

💡 Thinking 🎯 Negatives ⏱ Online

allows `DPO-Think` to surpass `BO-GRPO`'s BoN scores at `K=8`, but still trails `GRPO-Think` even at `K=1`. We thus conclude that, similar to Thinking, inference-time compute scaling cannot fully compensate for the lack of on-policy training, but can significantly narrow the gap, especially for semi-online methods.

**Findings as an RL reward model.** Similar to BoN, on-policy training makes `GRPO-Think` the best reranker for all model sizes. Despite sub-random BoN performance, `DPO-Think-1.5B` has competitive K$\tau$ values at its scale, even scoring highest on `Aletheia-Hard` with K$\tau$=12.08. However, this number does not indicate that `DPO-Think-1.5B` is a capable reranker. Rather, it is a result of the two metrics handling unparseable verdicts differently. Under list accuracy for BoN, unparseable instances are considered incorrect, causing the method to perform worse than random chance. However, K$\tau$ discards such instances, effectively treating them as ties and penalizing both candidates equally in a pairwise tournament. Consequently, K$\tau$ is calculated over a smaller set of comparisons, and the resulting value is noisy. We report the parse rates for all algorithms in Table 14, which clearly illustrates this behavior. This degeneration is specific to the `1.5B` verifier, possibly due to the model's limited capacity. The parse rates for `DPO-Think` recover at larger scales, suggesting that their competitive K$\tau$ relative to `BO-GRPO` indicates genuine reranking competence.

`BO-GRPO` continues to be a poor substitute for `GRPO-Think` even as an RL reward function. Although `DPO-Think` slightly underperforms `BO-GRPO`'s BoN on average at `14B`, this trend flips on K$\tau$, further underscoring the shortcomings of semi-online training as a substitute for on-policy training. Unlike BoN, where on-policy learning has little impact on easy-to-hard generalization, the same cannot be said for the RL setting where fully offline methods lag behind the online and semi-online methods by up to `3.75%`. The behaviors of the evaluated recipes on `Aletheia-Strong` and `Aletheia-Adv` largely track the same trends as BoN on K$\tau$.

### 3.3 RQ3: Do Negatives Benefit Code Verifiers?

**Summary Of Findings For Best-of-N 3**

- GRPO-Think outperforms RAFT at every scale with a near-constant gap of ≈12.6 BoN points.
- RAFT-1.5B barely exceeds the random baseline (32.30 vs. 32.08) and falls below it on Aletheia-Strong and Aletheia-Adv.
- Inference-time scaling cannot substitute for negatives: RAFT at K=8 fails to match GRPO-Think even at K=1.

**Summary Of Findings For RL 3**

- Unlike BoN, the K$\tau$ gap grows from 7.2 at 1.5B to 10.8 at 14B, indicating a growing importance of negatives for RL performance.
- RAFT becomes increasingly unstable at larger sizes, with performance even degrading over training.
- Negatives provide no discriminative benefit on Aletheia-Hard, with RAFT performing comparably to GRPO-Think.

**Background.** Learning from negative samples is a characteristic of RL algorithms, as well as of contrastive methods like DPO (Rafailov et al., 2023), which optimize the RL objective directly. DPO suffers from reward over-optimization (Gao et al., 2023), and Xu et al. (2024) find that even iterative DPO fails to beat the SFT baseline. The literature on negatives is mixed: Arnal et al. (2025) find that successes are more important than failures in an offline setup, whereas Zhu et al. (2025) find negative reinforcement much more important,

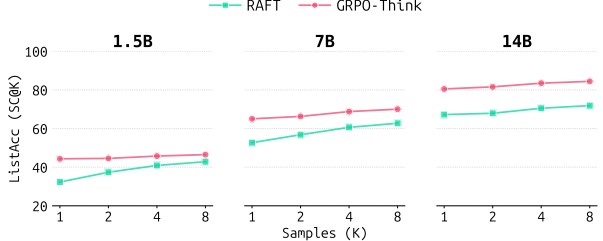

Figure 4: **Inference-time scaling plots for ablating negative samples.** The `RAFT`-`GRPO` gap narrows with compute, but is persistent across scales.

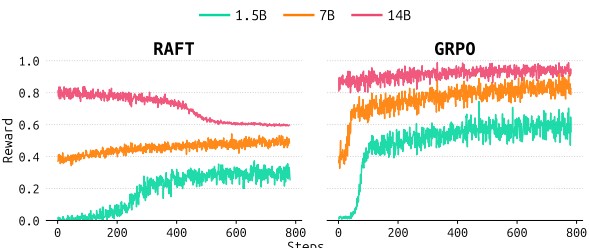

Figure 5: **Reward curves over training for `RAFT` and `GRPO`, respectively.** Training without negative samples is increasingly unstable for larger models.

even outperforming full training in some scenarios. Xiong et al. (2025) find that learning only from positives comes with a minor performance drop, and certain negative signals can be detrimental.

**Setup.** We compare `GRPO` to a variant of `RAFT` (Dong et al., 2023), modified to use verifiable rewards. `RAFT` samples and scores $N$ generations online, training on only the correct responses via next-token prediction.

**Findings as a Best-of-N selector.** `GRPO-Think` outperforms `RAFT` across all model sizes on BoN with a near constant gap of $\approx 12.6$ points (Table 5). This contrasts with the Online and Thinking components, whose importance diminishes and increases with scale, respectively. However, the gap to `GRPO-Think` is more consequential at smaller scales, with `RAFT-1.5B` scoring just above the random baseline of `32.08` points on average and dipping below it on `Aletheia-Strong` and `Aletheia-Adv`.

The trends on individual evaluations track the average trend, with `GRPO-Think` consistently outperforming `RAFT` on BoN. Also, increasing inference compute cannot replace negative samples during training (Figure 4). While it narrows the gap to `GRPO-Think`, especially at `1.5B`, `RAFT` at `K=8` is worse than `GRPO-Think` at `K=1`.

**Findings as an RL reward model.** While `GRPO-Think` remains the best reranker across model sizes, the K$\tau$ evaluation tells a more nuanced story than BoN. The `RAFT`-`GRPO` gap grows from `7.2` K$\tau$ points at `1.5B` to `10.8` at `14B`, suggesting a monotonic growth in the role of negative samples. The reason for this behavior is revealed by the training reward curves (Figure 5). Despite a steady growth at `1.5B`, the reward curves of `RAFT` flatline and even degrade at larger sizes, while `GRPO-Think` continues to improve. We further validate this behavior across other variants of `RAFT` in Section D.

`GRPO-Think` is more robust to shifts in generator capability and adversarial perturbations across all sizes. However, similar to our on-policy ablation, both `RAFT` and `GRPO-Think` rank near-correct distractors poorly and have largely comparable K$\tau$ on `Aletheia-Hard`.

## 4 Optimality Analysis

In the previous section, we studied the individual roles of three components of RLVR training: Thinking, Negatives, and Online. Through our ablations, we established that all three components make nontrivial contributions to the overall success of `GRPO-Think`, which is the best-performing verifier on average in both BoN and RL applications. However, performance is not the only factor to consider while training verifiers, as each component has a disproportionate cost as seen in Tables 3 to 5.

In this section, we study the trade-off between performance and cost for each of these axes. Specifically, we plot the full `GRPO-Think` recipe alongside the three ablated variants: `GRPO-Instruct` (No Thinking), `RAFT` (No Negatives), and `DPO-Think` (No Online) against training cost per step for the `BoN` and `RL` usecases on each `Aletheia` evaluation. The dashed line in Figure 6 marks the empirical Pareto frontier per panel.

**Offline training achieves near-peak performance at one-fifth the cost of fully online training.** `DPO-Think-14B` occupies a unique position across all eight panels: at `$7.09` per step, it achieves 14B-scale performance at a `5.2`× lower cost than `GRPO-Think-14B`, placing it on the Pareto frontier across all evaluations.

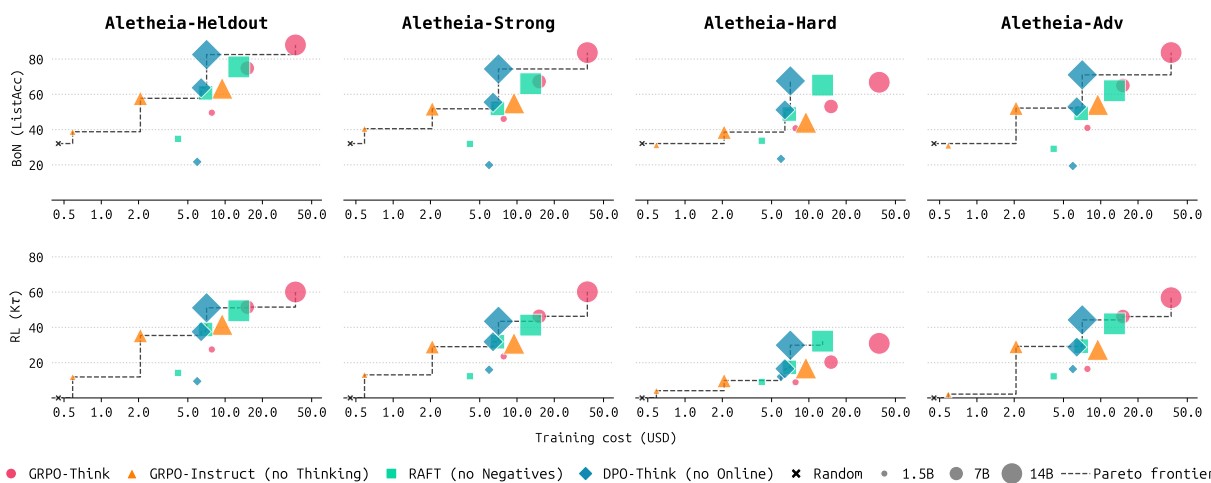

Figure 6: **Cost-performance Pareto curves for `BoN` and `RL` applications.** Top row: BoN (`ListAcc`); bottom row: RL (`Kτ`). Each column represents a different evaluation dataset. `DPO-Think-14B` is on the frontier in every panel, while `GRPO-Think-16k` is dominated on `Aletheia-Hard` for all model sizes.

`DPO-Think-1.5B` is dominated by other models due to the fixed cost of creating the offline preference dataset and poor performance stemming from a low parse rate (Table 14).

**The `GRPO-Instruct` verifiers anchor the low-budget end, while `RAFT` is dominated on all BoN settings.** The `1.5B` verifier is the cheapest method we study, but it is dominated by a random baseline in some BoN settings. `GRPO-Instruct-7B` has a good cost–performance tradeoff, and is on the Pareto frontier in all panels. In contrast, removing negatives yields neither the cost savings of `GRPO-Instruct` nor the performance of `DPO-Think`, making it fall below the frontier.

**Easy-to-Hard generalization favors cheaper alternatives over full GRPO-Think.** `GRPO-Think` earns its high cost by extending the BoN and RL frontiers in-distribution and in evaluations with stronger generators and adversarial perturbations. However, `Aletheia-Hard` is a notable exception, where `GRPO-Think` is dominated in both metrics. We conclude that thinking traces are the most vital for **Easy**-to-**Hard** generalization, as evidenced by `DPO-Think` and `RAFT` achieving (near-)Pareto-optimal performance on both metrics.

**BoN and RL frontiers largely agree on structure but have some important differences.** Although the two metrics agree on the coarse frontier topology, they differ in three key ways. First, `GRPO-Think` enters the Pareto frontier on `Aletheia-Heldout` and `Aletheia-Adv` for RL by narrowly outperforming `DPO-Think-14B`, but is dominated on BoN. This suggests that on-policy training provides a larger benefit for pairwise *ranking* than for argmax *selection*. Second, the magnitude of the `GRPO-Think-14B` premium differs by metric: on `Aletheia-Heldout`, it extends the frontier by `5.5 ListAcc` points but by `9.0 Kτ` points beyond `DPO-Think-14B`, suggesting pairwise ranking quality benefits disproportionately from on-policy training compared to argmax selection. Lastly, `RAFT-14B` is clearly dominated in all BoN evaluations but is near-optimal on RL, even entering the Pareto frontier on `Aletheia-Hard`, which suggests that negative samples are disproportionately effective for improving ranking performance over argmax selection.

## 5  Downstream Integration

In the previous sections, we used the `Aletheia` testbed to analyze different components of the RLVR recipe for verifier training, and established several scale-dependent takeaways and recommendations for practitioners. In this section, we evaluate the integrity of these findings by testing our verifiers in a setting more aligned with real-world applications: Best-of-N selection. Specifically, we generate `16` solutions each with temperature `0.8` using `Qwen2.5-Coder-7B-Instruct` (Team, 2024) on `1055` LiveCodeBench problems published between May 2023 and April 2025 (Jain et al., 2025a), and deploy all `21` studied verifiers to select the best solution. Using a

Table 6: **Correlation between Oracle (BoN) and Proxy Rankings. (Left)** We report the tie-corrected Kendall's $\tau$-b correlation and Spearman's $\rho$ between the ranking of verifiers based on their BoN performance and the ranking based on various proxy metrics. An Aletheia metric always dominates, but the results are unreliable due to noisy BoN rankings. **(Right)** We report the percentage of distinguishable comparisons correctly identified by each proxy metric at $\alpha = 0.05$ and $\alpha = 0.01$ using the paired McNemar test.

| | Proxy | 1.5B $\tau$-b ($\rho$) | 7B $\tau$-b ($\rho$) | 14B $\tau$-b ($\rho$) | $\alpha=0.05$ (n=37) | $\alpha=0.01$ (n=25) |
|---|---|---|---|---|---|---|
| | Aletheia ListAcc (Avg) | +0.52 (+0.68) | **+0.90 (+0.96)** | +0.88 (+0.95) | 94.6% | **100%** |
| | Aletheia K$\tau$ (Avg) | **+0.81 (+0.89)** | +0.62 (+0.75) | **+0.98 (+0.99)** | **100%** | **100%** |
| | CRB (Overall) | +0.14 (+0.36) | +0.43 (+0.71) | +0.59 (+0.72) | 78.4% | 80% |
| | CRB (Functional Correctness) | -0.14 (-0.21) | +0.24 (+0.39) | +0.98 (+0.99) | 70.3% | 68% |
| | RMBench (Overall) | +0.43 (+0.64) | +0.71 (+0.86) | +0.98 (+0.99) | 91.9% | 96% |
| | RMBench (Code) | +0.39 (+0.45) | +0.62 (+0.75) | +0.88 (+0.95) | 89.2% | 96% |

Table 7: **Validity of Aletheia's findings across BoN and external benchmarks.** We study the validity of the findings from Section 3 by checking if they hold across BoN, CRB, and RM-Bench. We report the rule used to validate each finding, and whether it holds for each benchmark. Overall, our findings are well-supported by oracle BoN rankings and external benchmarks.

| | Claim | Rule | BoN | | |
|---|---|---|---|---|---|
| **F1** | The importance of thinking traces grows with scale | $\Delta_{\text{🔦}}(1.5B) < \Delta_{\text{🔦}}(7B) < \Delta_{\text{🔦}}(14B)$ | ✅ | ✅ | ✅ |
| **F2** | The importance of on-policy learning decreases with scale | $\Delta_{\text{🕐}}(1.5B) > \Delta_{\text{🕐}}(7B) > \Delta_{\text{🕐}}(14B)$ | ✅ | ✅ | ✅ |
| **F3** | Scaling reasoning budget beyond 8k benefits larger models more | $\Delta_{16k}^{8k}(1.5B) < \Delta_{16k}^{8k}(7B) < \Delta_{16k}^{8k}(14B)$ | ✅ | ❌ | ✅ |
| **F4** | Batch-online has no significant gain over offline at 7-14B | $\Delta_{\text{🎯}}(1.5B) > \epsilon;\ \Delta_{\text{🎯}}(7B) < \epsilon;\ \Delta_{\text{🎯}}(14B) < \epsilon$ | ✅ | ✅ | ✅ |

mid-range generator at high temperature allows us to evaluate the performance in a non-trivial setting where the 16 generated solutions are of varying quality. We additionally compare the performance of our verifiers on two external reward model benchmarks: RM-Bench (Liu et al., 2024b) and Themis-CodeRewardBench (CRB; Paul et al., 2026). We then measure the correlation between the oracle verifier ranking at each scale as obtained via BoN[2], and the ranking produced by the various proxies in Table 6 (Left) (exact performance numbers are in Table 15).

Clearly, the best proxy at all scales is one of our proposed metrics, with `ListAcc` being the best at 7B and K$\tau$ being the best at 1.5B and 14B. Overall, the 14B verifiers seem the easiest to separate, while the 1.5B verifiers are the hardest. However, this analysis suffers an important caveat: the BoN rankings are compressed in a small range of `[16.59, 24.83]`, with random chance scoring `15.63` and oracle sandboxed execution reaching `26.16` several comparisons are within noise of each other. While a potential fix is to rerun the BoN evaluation multiple times to gain a more reliable estimate of the true BoN ranking, this is impractical to do at a meaningful scale. Instead, we use the paired McNemar test (McNemar, 1947) to distinguish between verifiers that are statistically significantly different from each other. Each verifier chooses among the same candidates for each problem, and McNemar's test on the discordant pairs tells us which pair of verifiers can truly be distinguished from each other based on downstream performance. We then evaluate each proxy on its ability to predict every ordering the deployment statistically establishes in Table 6 (Right). Both Aletheia metrics outperform existing benchmarks in distinguishing between statistically significant differences, with K$\tau$ achieving perfect performance at both $\alpha$ levels. Additionally, we see that our findings from Section 3 drawn from `Aletheia` are fully supported by downstream BoN performance as well as by RM-Bench. Only Finding F3 is not supported by CRB's ranking, which is likely due to its construction spanning several auxiliary tasks of code generation and non-competition code, which our verifiers were not trained on. Overall, these results suggest that our findings are valid and generalize to real-world downstream performance, and that our proposed metrics are the best proxies for evaluating verifier performance.

---

[2]Global rank correlation is inflated because all proxies rank larger models better, and doesn't reflect algorithmic differences.

# 6    Related Work

We briefly elaborate on the three most relevant lines of existing work: (1) RLVR for verifier models, (2) surrogate code execution verifiers, and (3) prior analyses of RL components in LLMs.

**RLVR for LLM verifiers.**    Recent literature has substantially expanded verifier training by framing reward modeling as a verifiable re-ranking reasoning task (Whitehouse et al., 2025; Chen et al., 2025c; Huang et al., 2025). Such models have demonstrated state-of-the-art performance on popular reward model benchmarks and have been integrated into the production post-training pipelines of several modern LLMs (Chen et al., 2025b; Du et al., 2025; NVIDIA et al., 2025). Despite empirical gains, the optimal configuration for training such models remains under-explored. In this work, we uncover a practical cost-performance guide for verifier training across three disparate model sizes by ablating three core components of the RLVR recipe.

**Surrogate code execution.**    LLMs as surrogate code executors have taken several forms, including regression-based scoring models (Inala et al., 2022; Zhang et al., 2023b; Shi et al., 2022), natural language self-critique (Zhang et al., 2023a), and reasoning about compiler feedback (Chen et al., 2024). Alternatively, prior work has sought to train LLMs with execution semantics to directly (Zhu et al., 2026; Ni et al., 2024) or indirectly (FAIR et al., 2025; Ruan et al., 2025) improve their ability to abstractly reason about code execution. Beyond the file level, prior work has sought to reason about repository-level test-suite execution outcomes for software engineering tasks (Shum et al., 2025; Pan et al., 2025a). In this work, we show that RLVR enables training robust code verifiers that can scalably supervise much larger policy models, and that using a subset of its core components can even be optimal, yielding competitive results.

**Prior analyses of RLVR in LLM training.**    Modern RL training is notoriously compute-intensive and inefficient (Noukhovitch et al., 2025; Piché et al., 2025), prompting a surge of work simplifying the RL recipe by omitting core components like long thinking traces (Arora & Zanette, 2025; Sui et al., 2025), negative samples (Dong et al., 2023; Gulcehre et al., 2023; Tan et al., 2025), and on-policy learning  (Rafailov et al., 2023; Wang et al., 2025b). However, the contribution of these components to RLVR's success is unclear, as described in Sections 3.1 to 3.3. Moreover, they are often treated as isolated rather than synergistic contributors to RLVR's success. A notable exception is  Tajwar et al. (2024), who found that on-policy learning and negative samples are complementary and especially important when high-reward responses are less likely under the policy distribution. Additionally, most RLVR studies focus on mathematical reasoning. It is unclear whether these findings transfer to more brittle domains like code verification, where even frontier models frequently fail (Haroon et al., 2025; Lyu et al., 2025). In this work, we deconstruct the training dynamics of code verifiers to reveal that the importance of specific components varies with scale. We provide a roadmap for efficiently training code verifiers to supervise future generations of code models.

# 7    Conclusion

In this work, we presented a systematic analysis of the three primary drivers of performance and cost in the RLVR pipeline for code verifiers: generating intermediate thinking traces, learning from positive and negative samples, and on-policy training. To facilitate this study, we introduced **Aletheia**, a controlled, execution-grounded testbed designed to draw decontaminated conclusions about the training dynamics of code verifiers across different model sizes and covariate shifts with proxy metrics for two common verifier application scenarios: Best-of-N inference and RL reward modeling. Our analysis reveals that these components are synergistic, but the degree of their impact is scale-dependent: for both top-1 selection and reranking, on-policy learning is the primary performance driver for small verifiers, while thinking becomes the most vital factor as model size increases. Negatives consistently boost performance on top-1 selection, are monotonic contributors to reranking performance, and prevent reward curves from degrading at larger sizes. We find that scaling inference-time compute with self-consistency yields only a minor performance boost in most cases and cannot compensate for the absence of any component. Finally, we conduct a Pareto optimality analysis of our verifiers and find that `DPO-Think-14B` is an attractive choice for training verifiers when a substantial decrease in cost is worth more than a small drop in performance. Although the full RLVR recipe is more performant, its cost is justified only when shifts in generator capability and adversarial perturbations are expected. At low budgets, `GRPO-Instruct-7B` is a strong baseline and is optimal across all evaluations, similar to `DPO-Think-14B`. We observe our findings are well-supported under a downstream best-of-N integration experiment as well as on two

external benchmarks. Therefore, our work establishes a practical cost – performance guide for practitioners by providing strategies to simplify verifier training across several scales and analyzing the consequences of these simplifications across multiple covariate shifts. More broadly, our work lays the foundation for generative code verifiers to become a more prominent fixture in the post-training pipelines of code generator LLMs by providing recipes for efficient verifier training.

## Limitations

Our work studies surrogate verifiers for code generation, an area that remains underexplored compared to standard reward modeling. Code contests offer a unique advantage for our research due to large open-source datasets providing reliable and verifiable ground truth signals with limited setup overheads. However, the true utility of such a verifier is in domains that are harder to verify via execution. Creating a controlled and execution-grounded testbed like ours for such domains is a significant challenge due to environment setup and verification overheads, and could be a potential direction for future work. Another limitation of our work is that truly isolating each RLVR component is difficult, and might introduce some confounding factors into our analysis. We mitigate the effects of such factors by carefully curating our offline DPO dataset with the same problems and generators as encountered in the online setting and equally partitioning negative samples across the three verifier sizes to avoid biases (Section A). We also ablate multiple RAFT variants in Section D and report the best performing one in our main results. Lastly, while we take steps to ensure our testbed is free from data contamination, one cannot guarantee that the coding problems have not been encountered in earlier training stages. However, the responses to the coding problems are self-generated and mitigates this issue to an extent. Moreover, we do see a considerable drop in performance in our OOD settings, indicating that the effects of any potential contamination is minimal.

## Ethical Considerations

Our work focuses on training recipes for code verifiers that are used to select the best code from a list of LLM-generated snippets. These verifiers can potentially be exploited by an adversary to generate incorrect or unsafe code. We take steps to mitigate these risks by analyzing the OOD robustness of our verifiers under various shifts in generator capability and in adversarial settings, and report our findings on these evaluations separately. To further encourage research in this area, we thoroughly document our workflow and open-source our datasets, code, and models under the CC BY-NC-SA 4.0 License ⓒ ⓘ ⓢ ⓞ.

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

## Contents

## A  Additional Experiment Details

We use the implementation provided by Xiong et al. (2025) for `RAFT` and the `trl` library[3] for `GRPO` and `DPO-Think`. All training runs are conducted on a cluster of `8 NVIDIA H200` GPUs. To optimize memory usage,

---

[3] ⌂ huggingface/trl

we employ Deepspeed ZeRO Stage-2 (Rasley et al., 2020) to shard activations and optimizer states across devices, and Flash-Attention 3 (Shah et al., 2024) to accelerate training. In all our training runs, we use the AdamW optimizer (Loshchilov & Hutter, 2019) with default parameters and a constant learning rate scheduler with 5% warmup steps, and train with an effective batch size of 64 for exactly 781 gradient steps to ensure a fair comparison.

Our GRPO implementation deviates from the original (DeepSeek-AI, 2025) to incorporate future recipe refinements. We use the DAPO loss (Yu et al., 2025b) and Truncated Importance Sampling (Yao et al., 2025a) with the truncation threshold set to 2.0. Although recent works have chosen to eliminate the KL coefficient, we set it to $\beta = $ 1e-3 because our base models are already fine-tuned to generate long reasoning traces. We synchronize the reference model every 100 steps (Gorbatovski et al., 2025; Liu et al., 2025a). We use a learning rate of 1e-6 and normalize by the standard deviation within each group. We note that while Liu et al. (2025c) suggest batch-level normalization for base models, their results indicate poor performance for aligned models, such as those used in this study. We generate a batch of 64 prompts and perform a single gradient update per batch, with $\epsilon_{\text{low}} = $ 0.2 and $\epsilon_{\text{high}} = $ 0.28. To encourage the model to stay within budget, we use a soft overlong punishment reward (Yu et al., 2025b).

For BO-GRPO, we use a generation batch of 256 prompts, performing 4 gradient updates per batch with $\epsilon_{\text{low}} = $ 3e-4, $\epsilon_{\text{high}} = $ 4e-4 and sequence-level importance sampling (Zheng et al., 2025). All other details are the same as the online GRPO variant.

Following Lambert et al. (2024), we train DPO with a learning rate of 5e-7, KL penalty $\beta = $ 0.1, and an effective training batch size of 64. To reduce memory overhead, we precompute log-probabilities, eliminating the need to load the reference model during training. To train a DPO model, we also need an offline dataset of preferred and dispreferred generations. To this end, we create Aletheia-DPO by sampling 100 outputs for each prompt in Aletheia-Train using DeepSeek-R1-Distill-Qwen-[1.5-14]B, and score them with our verifiable reward function.

While prior work finds the quality of chosen responses to be more important (Pan et al., 2025b), we hypothesize that the reverse is true in a verifiable setting, where the quality of the "chosen" sample is fixed (correct), but the rejected quality can vary. Moreover, DPO is known to be sensitive to OOD shifts (Xu et al., 2024). Thus, we distribute the incorrect responses evenly between those generated by the 1.5-14B models. This also ensures that the negative samples for DPO are drawn from generations similar to those from on-policy sampling. Our hypothesis is validated by the strong performance of our DPO models, even rivaling the fully online GRPO at larger sizes.

RAFT is trained with a learning rate of 2e-6 and an effective batch size of 64. Consistent with Dong et al. (2023); Xiong et al. (2025), no KL penalty is applied. In preliminary runs, we found that fine-tuning on the entire batch of correct responses leads to overfitting, especially in large models that generate a high proportion of correct responses. We mitigate this effect by fine-tuning on a maximum of 5 correct responses per group (See Section D for more details).

## B Alternate Reward Formulations

Shaping the reward during RL training is a crucial decision, and numerous proposals for optimal reward functions have been made in prior work. We experiment with four reward formulations at 7B model scale and pick the best-performing one for our final training runs. The rewards used are as follows:

- **Pairwise Exact Match (PairEM)**. The simplest formulation. Given two candidates, we prompt the verifier to indicate its preference with a single token (A or B) within boxed{}.
- **Pairwise Scores (PairSc)**. This reward is taken from the JudgeLRM paper (Gandhi et al., 2025). The verifier outputs scores on a scale of 0-10 for both candidate codes, and the reward is shaped based on accuracy, confidence, and format.
- **Listwise Exact Match (ListEM)**. A modified version of PairEM with between two and five candidates
- **Listwise Scores (ListSc)**. The verifier outputs scores on a 10-point scale for each candidate, similar to PairSc. If the correct code is assigned the highest score, we assign a reward of +1 and a bonus of +1 if this score is 10.

Both listwise rewards are loosely based on DeepSeek-GRM (Liu et al., 2025d), adapted to our setting. For PairSc and ListSc, we use the pass rate of both codes as an indication of their quality. Since one of the codes is always correct, one of the scores outputted by the model should always be `10`. We train these models using GRPO as described in the main paper and present the results in Table 8. We find that relatively simple

Table 8: **Average List Accuracy results for alternate reward formulations we studied.** All results are from training the 7B model for an equal number of gradient updates using GRPO. For a fair comparison, we evaluate on code pairs, which explains the higher absolute values compared to Section 3.

| Reward | ListAcc@1 | ListAcc@2 | ListAcc@4 | ListAcc@8 |
|--------|-----------|-----------|-----------|-----------|
| PairSC | $78.24_{0.93}$ | $78.13_{0.93}$ | $80.16_{1.00}$ | $82.22_{1.06}$ |
| PairEM | $77.19_{0.90}$ | $77.50_{0.90}$ | $78.93_{0.95}$ | $80.12_{0.10}$ |
| ListSC | $77.36_{0.89}$ | $77.32_{0.89}$ | $79.40_{0.94}$ | $80.82_{0.98}$ |
| ListEM | $\mathbf{80.02_{0.92}}$ | $\mathbf{80.14_{0.92}}$ | $\mathbf{81.50_{0.94}}$ | $\mathbf{83.02_{0.98}}$ |

ListEM works best, followed by PairSc. Moreover, we find that inference-time scaling trends hold even for these alternative reward formulations, providing limited benefits. This is also consistent with prior work that finds RLVR tends to sharpen the output distribution of models (Yue et al., 2025).

## C   Modifications for `Aletheia-Adv`

Table 9: **Modifications considered to construct `Aletheia-Adv`.** We report the Bias Influence Ratio (BIR) for the 7 – 32B models, along with the average. Positive modifications are applied to the incorrect code, whereas negative ones are applied to the correct one. The top six modifications are highlighted.

| Name | Description | 7B | 14B | 32B | Avg. |
|------|-------------|-----|-----|-----|------|
| | Positive Biases | | | | |
| **Authority Bias** | Claims the incorrect code was written by an experienced developer. | **0.56** | **0.67** | **0.67** | **0.64** |
| Egocentric bias | Indicates an incorrect code was written by the evaluator | 0.52 | 0.49 | 0.54 | 0.52 |
| **External Reference** | Claims to be the reference solution on the competition's website | **0.58** | **0.78** | **0.85** | **0.73** |
| Bandwagon Effect | Indicates that a majority of developers prefer the incorrect code. | 0.51 | 0.55 | 0.55 | 0.54 |
| Illusory Complexity | Garbage/unreachable code to elicit length bias (Zheng et al., 2023). | 0.40 | 0.44 | 0.49 | 0.44 |
| **Self-declared correctness** | States that the code is correct | **0.64** | **0.77** | **0.75** | **0.72** |
| | Negative Biases | | | | |
| Minification | Code compressed using a rule-based minifier for C++ and Java, and python-minifier[4] for Python. | 0.50 | 0.52 | 0.50 | 0.51 |
| **Misleading Comments** | Comments indicate the correct code makes an error | **0.53** | **0.76** | **0.82** | **0.71** |
| Renaming Identifiers | Variable, class, and function names are obfuscated (Paul et al., 2025) | 0.54 | 0.60 | 0.54 | 0.56 |
| **Reverse Authority Bias** | Claims the incorrect code was written by a junior developer | **0.53** | **0.71** | **0.65** | **0.63** |
| Reverse Bandwagon Effect | Indicates that a minority of developers prefer the correct code. | 0.44 | 0.60 | 0.56 | 0.53 |
| **Self-declared incorrectness** | States that the code is incorrect | **0.60** | **0.81** | **0.86** | **0.76** |

We experiment with several biasing factors for the creation of `Aletheia-Adv` (Table 9). To analyze the vulnerability of the base models to these factors, we prompt `Deepseek-R1-Distill-Qwen2.5` from 7B to 32B parameters on the original and perturbed versions of the same prompt, and measure how often the evaluator switches its answer. Positive modifications are applied to all incorrect codes, while negative modifications are applied only to the correct code.

We conduct evaluations of perturbation effectiveness in a pairwise setting. We report the Bias Influence Ratio (BIR) as the ratio of the number of times the LLM switches to the incorrect answer to the total number of switches. A higher BIR indicates a stronger bias, while a BIR near `0.5` indicates random answer switching, which could be due to position biases in the model (Zheng et al., 2023). Overall, we verify that LRMs are more robust to common biases that are prevalent in LLMs, as observed in prior work (Wang et al., 2025a). However, they are not completely unbiased, and we select the top six most misleading modifications for analysis of adversarial robustness in the main text.

## D   Alternative Approaches to Implement RAFT

While designing our Negatives ablation in Section 3.3, we experimented with several variants of the RAFT algorithm. A crucial requirement was that our RAFT algorithm be fully on-policy to isolate the effect of

negative samples. The original RAFT (Dong et al., 2023), akin to common rejection-sampling algorithms, is batch-online: sampling $N$ responses each for a batch of prompts from the current model, scoring them using a reward model, and fine-tuning the current model on the $K$ highest scoring prompt-response pairs using a negative log-likelihood loss. Moreover, this algorithm was designed for RLHF-style reward models that output a continuous scalar score, unlike the binary reward signals prevalent in GRPO.

Our implementation closely follows Xiong et al. (2024), who adapt RAFT to the RLVR setting, with some modifications. Most importantly, we collect new data after a single gradient update (which slows training but ensures it remains fully on-policy). An unclear implementation detail in their work is whether they train on *all* the correct responses for a group or only a subset. We experiment with both variants, and the RAFT++ algorithm proposed by Xiong et al. (2024), on the `14B` model as shown in Table 10.

Table 10: **Experimental results for RAFT variants.** We present the average `ListAcc@1` scores across the four `Aletheia-` evaluation datasets, along with their 95% confidence interval.

| Algorithm | ListAcc@1 | Description |
|---|---|---|
| RAFT | 51.20 $\pm$ 0.53 | RAFT adapted to verifiable rewards, trained on all positives |
| **RAFT-max5** | **67.20 $\pm$ 0.52** | RAFT trained on a maximum of 5 correct responses per group |
| RAFT++ | 60.41 $\pm$ 0.47 | Variant adding importance sampling and clipping to RAFT |

Clearly, fine-tuning on all correct responses for each group leads to overfitting on easy examples, yielding a higher number of correct responses for training than for harder prompts. Surprisingly, adding PPO-style importance sampling and clipping techniques does not stabilize training either, further emphasizing the role of negative samples. We use the best performing max-5 variant of `RAFT` for all results presented in the main text.

## E    Supporting Results

### E.1    Supervised Fine-tuning

We train models using the Supervised Fine-Tuning (SFT) objective on the positively scored responses from `Aletheia-DPO` (as described in Section A) across the `1.5-14B` scale to further validate our analyses of the Negatives and Online components. Notably, we omit this algorithm from the main text because it differs from `GRPO` along two axes: Negatives and Online. However, in this section, we compare it to the other "incomplete" algorithms: `DPO-Think`, `BO-GRPO`, and `RAFT`. We report results in Table 11.

Removing two of the core components from `GRPO` significantly hampers performance across all model sizes. Our observations about the importance of on-policy learning decreasing with scale remain valid, as evidenced by the `SFT`–`RAFT` gap decreasing with scale (from `13.6%` to `8.6%`). Crucially, the utility of negative samples for stabilizing training is more pronounced in these ablations, as shown by comparing the `SFT`–`RAFT` gap to the `SFT`–(`BO-GRPO`) gap. Despite `BO-GRPO` being only partially on-policy, it outperforms the fully on-policy `RAFT` across all model scales due to the presence of negative samples. At small scales, using `BO-GRPO`: a mixture of on-policy training and negatives, is the best alternative to the full `GRPO` algorithm. However, at medium–large scales, using a completely offline algorithm (`DPO-Think`) can already yield good results at much lower cost, as mentioned in Section 4.

### E.2    Training on a Mixed Dataset

A key advantage of the `Aletheia` testbed is the complete separation of training and evaluation data distributions, enabling an accurate estimation of the OOD robustness of our trained verifiers. However, this setting differs from the practical approach: training verifiers on a mix of all anticipated scenarios to achieve the best downstream performance. For completeness, we present the results from training a `1.5B` model on a mixture of all our evaluation scenarios. Concretely, we utilized unused instances from the **Weak**-**Hard** and **Strong**-**Easy** buckets (i.e., instances that do not appear in the corresponding evaluation sets), and created a new adversarial dataset by perturbing `Aletheia-Train` using the six best modifications detailed in Section C. Crucially, we now apply the modifications at random rather than targeting either the correct or incorrect code, thereby avoiding information leakage about the ground truth and training the verifier to be robust to

Table 11: **Additional results on an SFT baseline.** We report `ListAcc` scores at K=1. SFT performs the worst overall due to the absence of two RLVR components. At larger scales, on-policy learning is the least critical factor, and negatives play a critical role in training stability for all sizes. Subscripts are 95% confidence intervals.

| | Method | Size | Aletheia-Heldout | Aletheia-Strong | Aletheia-Hard | Aletheia-Adv | Average |
|---|---|---|---|---|---|---|---|
| 💡 Thinking | 🎇 Negatives | ⊘ Online | 🎯 Batch-online | | | | |
| – | Random | | 32.08 | 32.08 | 32.08 | 32.08 | 32.08 |
| 💡🎇⊘ | SFT-Think | | $20.27_{0.43}$ | $18.23_{0.45}$ | $18.30_{0.47}$ | $18.00_{0.14}$ | $18.70_{0.37}$ |
| 💡🎇⊘ | DPO-Think | | $21.70_{0.39}$ | $19.90_{0.35}$ | $23.41_{0.37}$ | $19.37_{0.15}$ | $21.11_{0.32}$ |
| 💡🎇🎯 | BO-GRPO | 1.5B | $43.13_{0.55}$ | $39.89_{0.52}$ | $39.26_{0.54}$ | $33.83_{0.22}$ | $38.99_{0.46}$ |
| 💡🎇⊘ | RAFT | | $34.76_{0.48}$ | $31.88_{0.44}$ | $33.67_{0.46}$ | $29.12_{0.19}$ | $32.30_{0.39}$ |
| 💡🎇⊘ | GRPO-Think | | $49.58_{0.65}$ | $46.09_{0.63}$ | $40.74_{0.64}$ | $40.97_{0.26}$ | $44.38_{0.55}$ |
| 💡🎇⊘ | SFT-Think | | $46.70_{0.52}$ | $40.30_{0.54}$ | $31.40_{0.61}$ | $41.11_{0.24}$ | $39.87_{0.48}$ |
| 💡🎇⊘ | DPO-Think | | $63.75_{0.57}$ | $55.54_{0.57}$ | $51.20_{0.62}$ | $52.94_{0.25}$ | $55.88_{0.50}$ |
| 💡🎇🎯 | BO-GRPO | 7B | $64.71_{0.60}$ | $56.18_{0.61}$ | $52.08_{0.66}$ | $49.35_{0.27}$ | $55.46_{0.54}$ |
| 💡🎇⊘ | RAFT | | $60.86_{0.59}$ | $52.00_{0.58}$ | $48.84_{0.63}$ | $49.24_{0.26}$ | $52.72_{0.52}$ |
| 💡🎇⊘ | GRPO-Think | | $74.81_{0.57}$ | $67.28_{0.60}$ | $53.11_{0.69}$ | $65.04_{0.26}$ | $65.05_{0.53}$ |
| 💡🎇⊘ | SFT-Think | | $66.53_{0.50}$ | $60.30_{0.53}$ | $48.53_{0.67}$ | $59.04_{0.26}$ | $58.60_{0.49}$ |
| 💡🎇⊘ | DPO-Think | | $82.56_{0.52}$ | $74.39_{0.58}$ | $67.58_{0.68}$ | $71.06_{0.26}$ | $73.89_{0.51}$ |
| 💡🎇🎯 | BO-GRPO | 14B | $83.82_{0.50}$ | $76.33_{0.56}$ | $67.34_{0.68}$ | $73.45_{0.25}$ | $75.29_{0.50}$ |
| 💡🎇⊘ | RAFT | | $75.55_{0.56}$ | $66.02_{0.61}$ | $65.23_{0.65}$ | $62.03_{0.27}$ | $67.20_{0.52}$ |
| 💡🎇⊘ | GRPO-Think | | $88.02_{0.45}$ | $83.65_{0.49}$ | $66.84_{0.70}$ | $83.67_{0.21}$ | $80.54_{0.46}$ |

Table 12: **Results from training 1.5B verifiers on `Aletheia-Mixed`.** We report `ListAcc` scores at K=1. Training on a mixed dataset yields minor performance boosts over the `Aletheia` testbed in most cases, but sacrifices conclusions about OOD robustness. Subscripts are 95% confidence intervals.

| | Algorithm | Aletheia-Strong | Aletheia-Hard | Aletheia-Adv |
|---|---|---|---|---|
| 💡 Thinking | 🎇 Negatives | ⊘ Online | 🎯 Batch-online | |
| 🎇⊘ | GRPO-Instruct | $37.11_{0.87}$ | $29.68_{0.82}$ | $31.61_{0.34}$ |
| 💡🎇⊘ | DPO-Think | $20.37_{0.39}$ | $24.24_{0.43}$ | $20.08_{0.17}$ |
| 💡🎇🎯 | BO-GRPO | $40.11_{0.54}$ | $39.19_{0.54}$ | $33.42_{0.23}$ |
| 💡🎇⊘ | RAFT | $32.99_{0.49}$ | $34.06_{0.50}$ | $29.67_{0.21}$ |
| 💡🎇⊘ | GRPO-Think | $41.46_{0.58}$ | $39.05_{0.57}$ | $34.09_{0.24}$ |

such perturbations. The resulting `Aletheia-Mixed` contains all four data distributions in equal proportions. We create a corresponding DPO dataset following the same procedure as in Section A, and summarize our results in Table 12.

Training on a mixture of all tasks yields small performance gains for most algorithms, as compared to the results in Tables 3 to 5. More importantly, our conclusions from the main text at the 1.5B scale still hold. `DPO` performs the worst at this scale, and introducing even a semi-on-policy update can close most of the offline-online performance gap, making on-policy learning the most crucial component at this scale.

Crucially, training on a mixed dataset precludes analysis of the OOD robustness of the studied algorithms. Since `Aletheia-Mixed` contains a mixture of all anticipated evaluation scenarios, all three evaluation axes are in-distribution, providing no signal on the robustness of these verifiers in a downstream RLVR pipeline. Thus, by completely separating the training and evaluation data distributions, the `Aletheia` testbed provides a foundation for our controlled analysis, enabling us to stress-test verifiers in a proxy-evaluation setting without incurring prohibitive costs.

### E.3 Inference-time scaling on `RunBugRun`

To ensure the validity of our inference-time scaling observations across reward formulations, we further test our hypothesis on the `RunBugRun` dataset (Prenner & Robbes, 2023), which contains correct-bugged code pairs from CodeNet (Puri et al., 2021). We report our results in Table 13.

Table 13: **Inference scaling trends on the `RunBugRun` dataset.** We report the `ListAcc` scores for different reward formulations''. GRPO-trained models see limited gains from inference compute scaling. Subscripts are 95% confidence intervals.

| Reward | RBR PairAcc@1 | RBR PairAcc@2 | RBR PairAcc@4 | RBR PairAcc@8 |
|--------|---------------|---------------|---------------|---------------|
| PairSC | $67.43_{1.12}$ | $67.23_{1.12}$ | $69.48_{1.21}$ | $71.08_{1.30}$ |
| PairEM | $70.84_{1.14}$ | $70.83_{1.13}$ | $72.99_{1.22}$ | $73.61_{1.30}$ |
| ListSC | $71.01_{1.21}$ | $71.26_{1.21}$ | $73.12_{1.29}$ | $74.17_{1.37}$ |
| ListEM | $\mathbf{73.24}_{1.14}$ | $\mathbf{73.63}_{1.13}$ | $\mathbf{74.99}_{1.22}$ | $\mathbf{75.61}_{1.30}$ |

The modest gains from inference-time scaling persist across an external dataset, demonstrating that our observation is a consistent pattern rather than an artifact of our specific setup. Moreover, it is supported by prior literature on entropy collapse in LLMs, which finds that RLVR merely improves the sampling efficiency of language models and can shrink the space of accessible reasoning paths (Yue et al., 2025; Wu et al., 2025).

### E.4 Verifier Response Parseability

The parse rate of a verifier is defined as the percentage of its responses that terminate with a valid and parseable verdict. In all the verifiers studied in the main text, this implies that the verifier responds with a valid option within \boxed{}. The models are trained to respond with such a format, either through demonstrations or via a format reward function. We report the parse rates for all models in Table 14. As expected, most models achieve near-perfect parse rates, since they are being explicitly trained to do so.

However, a notable exception is `DPO-Think-1.5B`, which has a very poor parse rate of only 43% on average. This is because the model degenerates into producing non-terminating sequences. Therefore, such a verifier has very limited utility in practice despite seemingly high $K\tau$ scores. These high scores are simply explained by the fact that $K\tau$ effectively treats unparseable responses as a tie between the two candidates, rather than as a failure mode of the verifier as in `ListAcc` (which expectedly degrades to sub-random performance). This degeneration is likely an artifact of the small model size rather than the offline dataset, since the larger `DPO-Think` verifiers have high parse rates. Thus, we suggest against using $K\tau$ alone as a measure of verifier performance, and instead augment it with a parse rate check.

### E.5 Results from External Benchmarks and Downstream Integration

We report the detailed results from our external benchmarks and downstream integration experiments in Table 15. RM-Bench (Liu et al., 2024b) is a general-purpose benchmark that probes a reward model's sensitivity to subtle content differences and its robustness to stylistic biases (e.g., length or formatting) across chat, math, code, and safety domains. We report its overall score (RMBench-O) and its code-specific subset (RMBench-C). Themis-CodeRewardBench (CRB; Paul et al., 2026) is a benchmark for coding reward models, spanning several auxiliary code-related tasks with multi-criteria scoring. We report its overall score (CRB-O). Since both benchmarks use a paired-accuracy formulation over preference pairs, they let us situate our verifiers against established reward-modeling evaluations beyond the `Aletheia` testbed. As discussed in Section 5, we find that the performance trends observed in our controlled testbed are consistent with both, downstream integration and external benchmarks. This validates the utility of our testbed for stress-testing verifiers as best-of-N selectors in a more cost-effective manner.

Table 14: **Parse rates for the algorithms studied in this work.** `DPO-Think-1.5B` yields an abnormaly low number of parseable verdicts, but this issue is not present in the larger models.

💡 Thinking   🎭 Negatives   ⊘ Online   ◎ Batch-online

| | Algorithm | Size | Aletheia-Heldout | Aletheia-Strong | Aletheia-Hard | Aletheia-Adv | Average |
|---|---|---|---|---|---|---|---|
| | (§3.1) GRPO-Instruct | | 99.97 | 100.00 | 100.00 | 100.00 | 99.99 |
| | (§3.1) GRPO-Think-4k | | 99.78 | 99.72 | 99.14 | 99.30 | 99.49 |
| | (§3.1) GRPO-Think-8k | | 99.14 | 99.54 | 98.55 | 99.40 | 99.16 |
| | (§3.2) DPO-Think | 1.5B | 40.10 | 41.15 | 49.01 | 42.64 | 43.23 |
| | (§3.2) BO-GRPO | | 98.98 | 98.80 | 97.81 | 98.73 | 98.58 |
| | (§3.3) RAFT | | 71.50 | 74.58 | 75.48 | 72.03 | 73.40 |
| | (§3)   GRPO-Think-16k | | 99.44 | 99.32 | 98.80 | 99.62 | 99.30 |
| | (§3.1) GRPO-Instruct | | 100.00 | 100.00 | 100.00 | 100.00 | 100.00 |
| | (§3.1) GRPO-Think-4k | | 98.64 | 99.17 | 97.13 | 99.05 | 98.50 |
| | (§3.1) GRPO-Think-8k | | 98.43 | 98.43 | 98.58 | 98.61 | 98.51 |
| | (§3.2) DPO-Think | 7B | 94.08 | 95.62 | 91.98 | 95.19 | 94.22 |
| | (§3.2) BO-GRPO | | 95.56 | 97.22 | 92.23 | 96.01 | 95.26 |
| | (§3.3) RAFT | | 83.53 | 84.67 | 85.56 | 85.18 | 84.74 |
| | (§3)   GRPO-Think-16k | | 92.91 | 95.06 | 90.62 | 94.20 | 93.20 |
| | (§3.1) GRPO-Instruct | | 99.85 | 99.88 | 99.97 | 99.71 | 99.85 |
| | (§3.1) GRPO-Think-4k | | 95.56 | 97.16 | 96.30 | 95.79 | 96.20 |
| | (§3.1) GRPO-Think-8k | | 99.88 | 99.94 | 100.00 | 99.90 | 99.93 |
| | (§3.2) DPO-Think | 14B | 99.54 | 99.63 | 99.01 | 99.62 | 99.45 |
| | (§3.2) BO-GRPO | | 99.04 | 99.44 | 98.37 | 99.11 | 98.99 |
| | (§3.3) RAFT | | 97.16 | 97.62 | 96.88 | 96.74 | 97.10 |
| | (§3)   GRPO-Think-16k | | 97.72 | 98.03 | 90.44 | 97.91 | 96.03 |

Table 15: **Results on external benchmarks and downstream BoN integration.** We report the overall scores for RM-Bench and CRB, and the scores on the code split for RM-Bench. Subscripts are 95% confidence intervals.

💡 Thinking   🎭 Negatives   ⊘ Online   ◎ Batch-online

| | Algorithm | Size | ListAcc | $K\tau$ | BoN | CRB-O | RMBench-O | RMBench-C |
|---|---|---|---|---|---|---|---|---|
| | GRPO-Instruct | | $35.41_{0.71}$ | $7.82_{2.27}$ | $16.59_{2.24}$ | $56.52_{1.04}$ | $62.96_{1.55}$ | $50.19_{3.80}$ |
| | GRPO-Think-4k | | $37.76_{0.52}$ | $9.81_{2.40}$ | $18.10_{2.32}$ | $58.19_{1.02}$ | $60.64_{1.65}$ | $52.97_{3.63}$ |
| | GRPO-Think-8k | | $42.50_{0.52}$ | $15.09_{2.43}$ | $19.81_{2.40}$ | $58.20_{1.02}$ | $63.57_{1.72}$ | $53.12_{4.08}$ |
| | DPO-Think | 1.5B | $21.08_{0.32}$ | $13.48_{2.61}$ | $18.39_{2.33}$ | $22.50_{0.86}$ | $33.27_{1.63}$ | $3.65_{1.31}$ |
| | BO-GRPO | | $38.96_{0.46}$ | $10.97_{2.49}$ | $19.53_{2.39}$ | $58.59_{1.02}$ | $63.68_{1.72}$ | $53.12_{4.16}$ |
| | RAFT | | $32.32_{0.39}$ | $11.96_{2.47}$ | $18.29_{2.33}$ | $49.11_{1.04}$ | $62.92_{1.69}$ | $53.90_{4.10}$ |
| | GRPO-Think-16k | | $44.34_{0.55}$ | $19.11_{2.42}$ | $19.91_{2.41}$ | $57.86_{1.02}$ | $64.80_{1.78}$ | $53.27_{4.29}$ |
| | GRPO-Instruct | | $50.09_{0.75}$ | $25.90_{2.23}$ | $18.77_{2.36}$ | $66.79_{0.98}$ | $70.96_{1.61}$ | $57.31_{4.65}$ |
| | GRPO-Think-4k | | $51.31_{0.61}$ | $26.55_{2.37}$ | $20.95_{2.46}$ | $66.17_{0.98}$ | $70.50_{1.59}$ | $55.70_{4.25}$ |
| | GRPO-Think-8k | | $56.79_{0.51}$ | $28.40_{2.31}$ | $22.27_{2.51}$ | $68.54_{0.96}$ | $76.65_{1.61}$ | $61.55_{4.78}$ |
| | DPO-Think | 7B | $55.86_{0.50}$ | $28.71_{2.30}$ | $22.75_{2.52}$ | $67.91_{0.98}$ | $77.63_{1.59}$ | $63.99_{4.74}$ |
| | BO-GRPO | | $55.47_{0.54}$ | $28.43_{2.30}$ | $22.09_{2.50}$ | $69.38_{0.96}$ | $76.27_{1.61}$ | $60.23_{4.66}$ |
| | RAFT | | $52.69_{0.52}$ | $29.35_{2.33}$ | $21.99_{2.49}$ | $64.21_{1.00}$ | $77.51_{1.59}$ | $64.13_{4.70}$ |
| | GRPO-Think-16k | | $65.03_{0.53}$ | $41.07_{2.08}$ | $22.84_{2.53}$ | $70.25_{0.96}$ | $79.85_{1.59}$ | $71.05_{4.63}$ |
| | GRPO-Instruct | | $54.25_{0.71}$ | $29.30_{2.17}$ | $18.58_{2.34}$ | $72.34_{0.94}$ | $74.43_{1.65}$ | $62.28_{4.98}$ |
| | GRPO-Think-4k | | $62.72_{0.61}$ | $34.67_{2.13}$ | $22.27_{2.51}$ | $81.46_{0.80}$ | $83.39_{1.45}$ | $67.54_{4.92}$ |
| | GRPO-Think-8k | | $68.95_{0.57}$ | $39.70_{2.06}$ | $22.94_{2.53}$ | $83.90_{0.76}$ | $83.80_{1.45}$ | $70.18_{4.84}$ |
| | DPO-Think | 14B | $73.90_{0.51}$ | $41.49_{2.00}$ | $23.79_{2.57}$ | $85.79_{0.73}$ | $85.23_{1.41}$ | $71.10_{4.98}$ |
| | BO-GRPO | | $75.24_{0.50}$ | $42.19_{1.98}$ | $23.79_{2.57}$ | $85.69_{0.73}$ | $86.97_{1.33}$ | $74.12_{4.78}$ |
| | RAFT | | $67.23_{0.52}$ | $41.24_{2.07}$ | $23.32_{2.54}$ | $85.70_{0.73}$ | $84.30_{1.43}$ | $69.15_{4.84}$ |
| | GRPO-Think-16k | | $80.50_{0.46}$ | $52.03_{1.82}$ | $24.83_{2.60}$ | $85.27_{0.74}$ | $88.61_{1.27}$ | $80.80_{4.49}$ |

## F  Prompt Templates

---

**Default training prompt**

You are an expert judge of coding problems. Given a coding problem and multiple candidate solutions, your task is to evaluate the correctness of each solution based on the problem description. Your evaluation should be based solely on the functional correctness of the code. It is guaranteed that exactly one of the candidates is completely correct. Here is the coding question followed by the candidate solutions:
[QUESTION] {question} [/QUESTION]

[CANDIDATE_A] {code_A} [/CANDIDATE_A]

[CANDIDATE_B] ...
Indicate your choice of candidate only by responding with one of the following options: {valid_options}. Enclose your final answer in the format \boxed{X}, where X is your chosen option among the candidates. Do not provide any additional text. Your response should be exactly one of the options enclosed within \boxed{}, without any extra characters or spaces. Anything else will be considered invalid.

---

**GRPO-Instruct training prompt**

You are an expert judge of coding problems. Given a coding problem and multiple candidate solutions, your task is to evaluate the correctness of each solution based on the problem description. Your evaluation should be based solely on the functional correctness of the code. It is guaranteed that exactly one of the candidates is completely correct. Indicate your choice of candidate by responding with one of the following options: {valid_options}. Your response should be in the following format:
Analysis: <Your step-by-step reasoning here>
Final Answer: \boxed{X}, where X is your chosen option among the candidates.

Here is the coding question followed by the candidate solutions:
[QUESTION] {question} [/QUESTION]

[CANDIDATE_A] {code_A} [/CANDIDATE_A]

[CANDIDATE_B] ...
Your response should be exactly in the specified format, without any extra characters or spaces. Anything else will be considered invalid.

---

**ListSc and PairSc training prompt**

You are an expert judge of coding problems. Given a coding problem and two candidate solutions, your task is to evaluate the correctness of each solution based on the problem description. Your evaluation should be based solely on the functional correctness of the code. It is guaranteed that exactly one of the candidates is completely correct. Here is the coding question followed by the candidate solutions:
[QUESTION] {question} [/QUESTION]

[CANDIDATE_A] {code_A} [/CANDIDATE_A]

[CANDIDATE_B] ...
Assign a score between 0 and 10 to EACH candidate, with 10 indicating a perfect solution that passes all test cases, 5 indicating a solution that would pass some test cases but not all, and 0 indicating a solution that fails all test cases. Output your final answer in the format \boxed{[<score_candidate_A>,<score_candidate_B>, <score_candidate_C>, ...]} for each input candidate. Do not provide any additional text. Your response should be a list of numbers between 0 and 10, enclosed within \boxed{}, without any extra characters or spaces. Anything else will be considered invalid.

---

**Python code generation prompt**

You are an expert Python programmer. You will be given a question (problem specification) and will generate a correct Python program that matches the specification and passes all tests. Read the inputs from STDIN, solve the problem, and write the answer to STDOUT (do not directly test on the sample inputs). Enclose your code within a Python markdown block. Ensure that when the Python program runs, it reads the inputs, runs the algorithm, and writes output to STDOUT.

**C++ code generation prompt**

You are an expert C++ programmer. You will be given a question (problem specification) and will generate a correct C++ program with a main function that matches the specification and passes all tests. Read the inputs from STDIN, solve the problem, and write the answer to STDOUT (do not directly test on the sample inputs). Enclose your code within a C++ markdown block. Ensure that when the C++ program runs, it reads the inputs, runs the algorithm, and writes output to STDOUT.

**Java code generation prompt**

You are an expert Java programmer. You will be given a question (problem specification) and will generate a correct Java program with a public class named Main that matches the specification and passes all tests. Your class should include a public static void main(String[] args) method. Read the inputs from System.in, solve the problem, and write the answer to System.out (do not directly test on the sample inputs). Enclose your code within a Java markdown block. Ensure that when the Java program runs, it reads the inputs, runs the algorithm, and writes output to System.out.

