# OpenReview forum: "Aletheia: What Makes RLVR For Code Verifiers Tick?"
_TMLR — Under review for TMLR_

### Review · Reviewer_Xr9a · 2026-06-25

**Summary Of Contributions:**

The paper considers the problem of post-training models for coding tasks. We have a "generator" model that generates code based on prompts. We'd like to score the generator's code, and use the score for training the generator via reinforcement learning. On well-defined problems (e.g., coding problems with tests), we can easily just run the code. But, for more complex and/or ambiguous tasks, it's difficult to evaluate the code. Instead, we use "verifier" models that process the code and return a verdict. The idea is that the verifier model will learn a preference for higher quality code that generalizes beyond easy-to-check coding problems.

There's a popular approach for training verifiers called "GRPO", which includes using 1) reasoning chains, 2) positive and negative examples, and 3) online training. The goal in this paper is to ablate each of these three techniques across models with 1.5b, 7b, and 14b parameters. The challenge is that evaluating a verifier requires actually training a generator, which can take something like 1500 hours until the metric isn't just noice and depends on a lot of other factors. The paper cites Kim et al. 2025 and Wen et al. 2025 to justify using the following two metrics as *proxies* for the verifier's quality:
* BoN = 1 if model successfully identifies the best of several candidate code solutions and 0 else,
* RL = Kendall's $\tau$ as the fraction of candidate code solutions that are in the correct relative rank.

The paper creates a dataset of code solutions for roughly 4k coding questions. Each code solution is run against roughly 25 tests, and scored based on how many tests it passes. The solutions are then partitioned into several groups based on how many code solutions successfully pass the tests and how similar the candidate solutions are, among other factors. The paper ablates the three techniques and reports the two metrics across the four groups of code solutions, for each of the three model sizes. They then summarize the results. My TL;DR takeaway is that all three techniques help (Tables 3, 4, and 5).

**Audience:**

Yes

**Audience Explanation:**

RL for post-training is clearly important and, by extension, so is training good verifiers. I think people would be interested in this paper, but I think the findings aren't meaningful because of the structural issues I mention above.

**Claims And Evidence:**

No

**Claims Explanation:**

At a more minor level, I think the authors should add experiments for larger models (14b is tiny by today's standards) and report confidence intervals in the main body (the metrics are reported to 4 significant figures, but there's no way that the last digits aren't noise).

At a higher level, I think there are two structural problems:

1. The whole argument for using a verifier is that there's some code that cannot be easily evaluated (e.g., it's expensive to run or it's underspecified). However, the verifiers in this paper are evaluated on code that's very easy to run and is well-specified. This makes the results much less interesting because they quantify verifier performance in the "toy" scenario rather than the "real world". I think it would be much more meaningful if the verifier's were evaluated on more real world tasks: The authors don't even have to come up with these, they can use existing strong benchmarks.

2. A more concerning issue is the validity of the two metrics themselves. The paper cites Kim et al. 2025 and Wen et al. 2025 to justify using BoN and Kendall's $\tau$ as a proxy for verifier performance on training generators. However, the setup of those works and this paper is different: there, the verifier models produce a single number via a linear head whereas here, the verifier can reason and outputs a "boxed" verdict. More concerningly, neither paper actually concludes that BoN or Kendall's $\tau$ is a good metric for downstream verifier performance: Wen et al. 2025 actually find that that accuracy (measured via BoN or Kendall's or like five other methods) is not well-correlated with downstream verifier performance (see Table 4). Kim et al. 2025 study the "overparameterization" of generators to verifier reward, and I think (?) don't weigh in on metrics for verifier performance besides talking about it through the lens of their $\gamma$ overparameterization factor. Given that the two metrics used in this work aren't shown to indicate downstream verifier performance, I don't see how the results are meaningful.

The only solution I see to accurately evaluate the verifier is to actually use the verifiers to train a generator and report performance with the various ablations. I know it would be computationally intensive but that's kind of the name of the game when doing LLM training research. More broadly, though, I think the findings of this paper are kinda obvious (all three techniques help for verifier training) so maybe it's worth reorienting the contribution: it would be *really* nice if the authors could find a metric for verifiers that *does* accurately predict downstream performance (though I guess Wen et al. 2025 tried to do this and concluded they couldn't find anything).

**Requested Changes:**

Requested changes:

[critical] Larger models

[critical] Confidence intervals in main body

[critical] More realistic evaluation sets (ideally, using someone else's benchmark)

[critical] Downstream evaluation of verifier performance

---

> ### Author Response · Authors · 2026-07-10
> **Response to reviewer Xr9a (1/3)**
>
> We thank the reviewer for the detailed feedback on our work and for highlighting its relevance to the community. We have revised our manuscript in accordance with the requested changes, which we describe here as follows.
>
> **1. Experimental integrity**
> We appreciate the reviewer’s feedback in this area. We respond to this concern in two subsections.
>
> **1.1 Validity of the metrics used**
> We first clear up the confusion regarding our use of Wen et al. (2025) and Kim et al. (2025) in our work. We construct Aletheia as a reliable reward model benchmark in accordance with Kim et al. (2025)’s findings, as follows:
> (i) minimize differences between chosen and rejected responses beyond correctness: Within each prompt, responses are sampled from the same model. Aletheia-Hard further reduces the difference between responses through near-correct distractors.
> (ii) multiple comparisons across a wide range of chosen and rejected responses: lists of length 2–5 codes.
> (iii) responses should be sourced from a variety of models: multiple weak and strong generators used across different prompts.
>
> Our choice of metrics is informed by Wen et al. (2025). They find, "Increasing the number of responses per prompt can enhance the correlation between measured RM error and policy regret" (Finding 4), and Kendall’s tau has a stronger correlation with downstream performance than paired accuracy (Table 4). However, the reviewer is correct that these findings are based on scalar RMs, which differ from the GenRMs in this paper.
> We have accordingly corrected and softened Section 2.2 as follows:
>
> > (Old) Despite **having high correlation with BoN**, accuracy alone is insufficient to predict the utility of a verifier as an RL reward-model (Kim et al., 2025). [...] Thus, we evaluate our verifiers' ability to reconstruct the full ranking of N candidates rather than just top-1 selection accuracy, **which is proven to better predict** downstream RL performance (Kim et al., 2025; Wen et al., 2025; Feng et al., 2025).
> > (New) However, **accuracy alone is often insufficient** to predict the utility of a verifier as an RL reward-model (Wen et al., 2025; Feng et al., 2025). [...] Thus, we evaluate our verifiers’ ability to reconstruct the full ranked order of N candidates rather than selecting the best candidate, **which may better predict** downstream RL performance (Wen et al., 2025).
>
> **1.2 Validation on external benchmarks and downstream Best-of-N**
> In accordance with the reviewer’s suggestion, we have uploaded a revised manuscript containing two additional benchmarks: RM-Bench (Liu et al., 2025) and CodeRewardBench (CRB; Paul et al., 2026). We also deploy our verifiers as Best-of-N (N=16) selectors with Qwen2.5-Coder-7B-Instruct as the generator on LiveCodeBench v1-v6 (1055 problems). We compare each verifier’s BoN selection accuracy to the oracle best-of-16 obtained via sandboxed execution. We evaluate all 21 verifiers on these benchmarks and report the rank correlations between their produced rankings and Aletheia’s. We will provide a TL;DR of the results here. For detailed results, we kindly ask the reviewer to read Section 5 of our revised manuscript.
>
> **TL;DR:** Aletheia’s metrics are better correlated with verifier rankings determined by downstream Best-of-N selection. To avoid drawing conclusions from noisy BoN point estimates, we run paired McNemar tests over the shared problem set to identify the verifier pairs whose BoN difference is statistically resolved. Aletheia achieves the highest agreement with BoN of any benchmark, ordering at least 35 of the 37 pairs resolved with 95% confidence correctly, and all 25 of those resolved with 99% confidence (Table 6). Moreover, we observe that all our findings in Section 3 hold under the oracle Best-of-N verifier ranking, but some cannot be supported through external benchmarks alone (Table 7). Taken together, these findings highlight the importance of a controlled evaluation testbed for verifier training. Unfortunately, downstream RL integration isn’t a feasible experiment for us to conduct with the resources at hand due to issues with training instability and cost, as mentioned in Section 1 of our manuscript. Thus, we have modified our manuscript to appropriately hedge our claims about downstream RL transfer.
>
> **2. Add confidence intervals in the main text**
> We thank the reviewer for their suggestion. We have updated all tables in the main text and appendices to include 95% confidence intervals, calculated over 16 generations per prompt, at a temperature of 0.6 and a top-p of 0.95. This change is reflected in the updated manuscript.

---

> > ### Author Response · Authors · 2026-07-10
> > **Response to reviewer Xr9a (2/3)**
> >
> > **3. Experiments for larger models**
> > We appreciate the opportunity to clarify this point. While modern LLMs have far outgrown the models studied in our work, verifiers often require much fewer parameters than the generators they supervise. This is likely because verifying a solution is easier than generating one from scratch (Wei, 2025): a hypothesis that has been empirically validated for LLMs (Kenton et al., 2024; Song et al., 2025; Xiang et al., 2025), and is largely the motivation behind the scalable oversight problem (Christiano et al., 2018; Bowman et al., 2022). While 14B verifiers are indeed "tiny" compared to trillion-scale LLMs, verifiers are often sub-32B models (Paul et al., 2026; Malik et al., 2026; Shum et al., 2026; Zhu et al., 2026), which suggests that our verifiers appropriately capture the current post-training ecosystem. Additionally, our new BoN integration results from Section 5 of the updated manuscript clearly show that our verifiers cover the full spread between random chance and oracle performance, with GRPO-Think-14B-16k reaching 94.9% of oracle performance. Thus, further scaling would likely not help much.
> >
> > **4. Evaluation on a toy setting**
> > We thank the reviewer for this point, which allows us to explain our design choice. We agree that a verifier’s ultimate value lies in settings where execution is costly. However, we would push back on the reading that this makes our evaluation a "toy scenario". While the codes in our dataset are easy to label, their surrogate verification is not necessarily easy. Our hardest evaluation set contains near-correct distractors (Aletheia-Hard), and every model size shows a large drop in accuracy on it, with even the 14B verifiers reaching <70% list accuracy. Thus, our studied scenarios are a demanding verification problem despite the ground truth being cheap to obtain.
> >
> > Reliably evaluating a verifier's competence requires an objective correctness label, and execution against test cases is the only such signal for code. Evaluating instead on underspecified or inexecutable code would introduce circularity, with one LLM judging another, and weaken our findings. Real-world software-engineering tasks are verifiable in principle, but constructing an equivalent controlled testbed is substantially harder, as every difficulty we already face in executing competition-level code is exacerbated, thereby placing them out of scope for this work. Nonetheless, we concede this as a limitation of our work, and a promising direction for future research. We have added a Limitations section to make this explicit as follows.
> >
> > > Our work focuses on verifiers for competitive programming, an area that remains underexplored compared to standard reward modeling, as described in Section 1. Code contests offer a unique advantage for our research by providing a verifiable ground truth. However, the true utility of a verifier is in domains that are harder to verify via execution. Creating a controlled and execution-grounded testbed like ours for such domains is a significant challenge due to environment setup and verification overheads, and could be a potential direction for future work.
> >
> >
> > **5. Unsurprising findings**
> > We respectfully disagree with the reviewer’s reading of our findings. Their TL;DR takeaway is a reductive understanding of our contributions. In addition to establishing the collective importance of the three components, we unpack the importance of each component across the scaling spectrum, finding that on-policy learning is vital for small verifiers, whereas thinking traces are most crucial for larger ones. We also find that negative samples play a constant role in stabilizing training, and that inference-time scaling through self-consistency cannot compensate for any single component. Our optimality analysis in Section 4 additionally finds that offline training can achieve parity with fully online training at one-fifth the cost. Thus, our analysis goes beyond a simple ablation of components and uncovers previously unknown dynamics in verifier training.

---

> > > ### Author Response · Authors · 2026-07-10
> > > **Response to reviewer Xr9a (3/3)**
> > >
> > > **References**
> > > 1. Bowman et al.: Measuring progress on scalable oversight for large language models. arXiv 2022.
> > > 1. Christiano et al.: Supervising strong learners by amplifying weak experts. arXiv 2018.
> > > 1. Kenton et al.: On scalable oversight with weak LLMs judging strong LLMs. NeurIPS 2024.
> > > 1. Kim et al.: Rethinking reward model evaluation through the lens of reward overoptimization. ACL 2025.
> > > 1. Liu et al.: RM-Bench: Benchmarking Reward Models of Language Models with Subtlety and Style. ICLR 2025.
> > > 1. Malik et al.: Rewardbench 2: Advancing reward model evaluation. ICLR 2026.
> > > 1. Paul et al.: Themis: Training Robust Multilingual Code Reward Models for Flexible Multi-Criteria Scoring. arXiv 2026.
> > > 1. Shum et al.: SWE-RM: Execution-free Feedback For Software Engineering Agents. ICLR 2026.
> > > 1. Song et al.: Mind the Gap: Examining the Self-Improvement Capabilities of Large Language Models. ICLR 2025.
> > > 1. Wei: The asymmetry of verification and verifier’s law. 2025.
> > > 1. Wen et al.: Rethinking Reward Model Evaluation: Are We Barking up the Wrong Tree? ICLR 2025.
> > > 1. Xiang et al.: Towards system 2 reasoning in LLMs: Learning how to think with meta chain-of-thought. arXiv 2025.
> > > 1. Zhu et al.: CodeScaler: Scaling Code LLM Training and Test-Time Inference via Reward Models. arXiv 2026.

---

> > > > ### Comment · Reviewer_Xr9a · 2026-07-21
> > > >
> > > > The first author isn't the only one who contributed, please make acknowledging *all* authors standard!

---

> > > ### Comment · Reviewer_Xr9a · 2026-07-21
> > >
> > > I thank the authors for their response. I think of the rebuttal as a chance to take the reviewer's feedback to heart or, if it's wrong, point out why.
> > >
> > > I appreciate that the authors added the additional benchmarks and uncertainty (side note: if there are only 16 runs, what is the 95th percentile?).
> > >
> > > However, they pushed back on the remaining requests and points, notably the validity of the metrics and obviousness of the findings. Personally, I found their responses unpersuasive; I still (A) read prior work as saying the metrics aren't useful for downstream performance and (B) think it's obvious that each of the three GRPO steps help.

---

### Review · Reviewer_yrvP · 2026-07-01

**Summary Of Contributions:**

This paper studies how to train generative code verifiers using Reinforcement Learning with Verifiable Rewards (RLVR). The main motivation being that executing code at scale is expensive and noisy and so learning some sort of a “surrogate” verifier is useful. A surrogate verifier meaning a model that takes as input a programming problem and some candidate code solutions, and then tries to either identify the best or rank the candidates, but without executing any code. The authors propose a testbed  for evaluating code verifiers and call it Aletheia. The benchmark includes an in-distribution heldout set as well as three out of distribution datasets, each with specific distributional shifts: candidates generated by stronger models, harder near-correct incorrect candidates, and adversarially modified parts of the code.  With this, the authors try to investigate which components of the full RLVR verifier-training recipe matter at which stage, the  components being: long intermediate reasoning traces, learning from negative samples, and on-policy training.

In terms of contributions, this benchmark / testbed itself is the main contribution, the introduction of such a benchmark can help understand RVLR better, and the second key contribution is precisely that. The authors use this testbed to do many carefully designed experiments to try and understand which RVLR parts are relevant in which kind of setting, contributing a cost-benefit analysis of each component.

The main strength of the paper is that the experiments are quite comprehensive, done at various scales, with various ablations and seeds, along with various metrics.
Additionally the paper also tries to present this rich set of results in an appealing manner with the use of intermediate summaries sprinkled throughout the paper to help the read not get lost in the overall details better.

At the same time, the main weakness I felt was that the authors were over-claiming in some places, and their interpretations of their experiments might have been a bit more enthusiastic, and while reading the paper, I thought of some places which probably require some hedging. I will put some more details about what I mean in the next section.

**Audience:**

Yes

**Audience Explanation:**

Yes, this is a very relevant topic, as LLMs are being quite widely adopted in coding, and so I believe many individuals in the TMLR audience, and in general the ML community would find this interesting.

**Broader Impact Concerns:**

No additional concerns.

**Claims And Evidence:**

Yes

**Claims Explanation:**

Broadly speaking yes, but I think it's a more qualified yes, because I would like some claims to be a bit more softened or stated more carefully.
For the high level empirical trends, there is fairly convincing evidence.
The benchmark is clearly designed for code verifiers, evaluation is done across multiple model sizes, metrics, and distribution shifts. The results in various experiments throughout the paper support the claim of the paper that full GRPO-Think (without ablations) is usually the strongest method in raw performance. The fact that that long reasoning traces become more useful at larger model scales is well supported by experiments in Section 3.1, that offline DPO-style training becomes more competitive at larger scales is well supported in Section 3.2, and that positive-only RAFT-style training generally underperforms without the use of negative samples in Section 3.3. The cost-performance analysis stating that DPO-Think-14B can be a strong Pareto tradeoff relative to the full online GRPO recipe is also supported by experiments in Section 4.


However, I do think that not all of these claims are supported equally well. Isolating individual components of RLVR, such as Thinking, Negatives, and Online training is really hard and in this paper as well, my reading is that, this is partially true. Several ablations compare the end up changing more than just one variable. For example in the DPO vs GRPO (Section 3.2), this is no longer just online vs offline training and the optimization objective that changes, but the data construction also changes.

Secondly, my reading was that some results are more mixed than the high-level framing. Different metrics sometimes favor different methods, and the conclusions vary depending on whether the verifier is evaluated as a Best-of-N selector or as a reward-model-style ranker. The Aletheia-Hard for instance is a case where strongest methods also do not have good performance and here some of the claims therefore do not hold as clearly. So the paper does do a good job in identifying some practical tradeoffs and limitations in current verifier training methods, but I would try to tone down the isolation of all components claim and some of the the blanket statements in the introduction, abstract and conclusion about certain things being better vs others.

Finally, the setting chosen for the code verifier is synthetic, for instance it does not cover the case properly when multiple things are correct / alternate ways of achieving the same results (tie in rankings), so the statement "our work provides the empirical foundation necessary to efficiently deploy robust code verifiers", seems a bit far fetched.

Another such claim for example is that the benchmark is contamination free. From what I understood that this is contamination free from the point of view of RL, but in pretraining such problems along with their solutions might have been seen. Granted as the authors are generating some of the candidates themselves, all solutions might not have this, but the question and the base candidate answer might have been.

Having said all of this, I would still lean towards “Yes” rather than no, and simply encourage the authors to give the paper another pass to soften some of the stronger claims in the paper.

**Requested Changes:**

The main changes I would like to propose are in terms of writing, I do think that the experiments are comprehensive. I have mentioned these already in my answers above. I am leaning towards an accept already, and would be willing to engage with the authors if they can defend some of these claims. In case, the authors are unable to do so, and do end up agreeing that some of their claims are strong, then yes, I would only want to recommend acceptance after the authors hedge and soften some of the stated claims.

---

> ### Author Response · Authors · 2026-07-10
> **Response to reviewer yrvP (1/1)**
>
> We thank the reviewer for their positive review and for recognizing the comprehensiveness of our experiments. We summarize the requested changes, now reflected in our revision, below.
>
> **1. Acknowledge difficulty of isolating components**
> We agree that different objectives may introduce confounding factors. We mitigate these where possible, e.g., by sourcing DPO responses from DeepSeek-R1-Distill-Qwen-[1.5-14B] on the same problems as Aletheia-Train. Using the same base models controls for stylistic preferences the DPO model might otherwise learn, and we equally partition the DPO "rejected" samples across the three sizes to avoid biasing the verifier toward generator size. While we have taken all steps possible to mitigate such risks, we concede this may be a limitation, which we now acknowledge in a Limitations section:
>
> > Another limitation of our work is that truly isolating each RLVR component is difficult, and might introduce some confounding factors into our analysis. We mitigate the effects of such factors by carefully curating our offline DPO dataset with the same problems and generators as encountered in the online setting and equally partitioning negative samples across the three verifier sizes to avoid biases (Section A). We also ablate multiple RAFT variants in Section D and report the best-performing one in our main results.
>
> **2. Synthetically constructed testbed**
> While our testbed does construct a synthetic scenario, we have added a downstream Best-of-N integration that addresses this concern. We deploy all verifiers as Best-of-N (N=16) selectors for a Qwen2.5-Coder-7B-Instruct generator and compare against the oracle Best-of-N obtained via sandboxed execution (Section 5 of the revision). The verifier rankings on Aletheia are well supported by this downstream application, where more complex situations such as ranking ties arise. We have also softened the statement quoted by the reviewer to avoid overclaiming (kindly refer to point 4).
>
> **3. Cannot guarantee true decontamination**
> We agree. Since model developers do not disclose pretraining data, the models might contain problem statements and some sample solutions. However, as the reviewer notes, the responses to the coding problems are self-generated, which somewhat mitigates this. Moreover, the considerable performance drop in our out-of-distribution evaluations indicates that even if these problems were seen during earlier training stages, they do not influence performance much. We nonetheless acknowledge this as a limitation:
>
> > While we take steps to ensure our testbed is free from data contamination, one cannot guarantee that the coding problems have not been encountered in earlier training stages. However, the responses to the coding problems are self-generated, which mitigates this issue to an extent. Moreover, we do see a considerable drop in performance in our OOD settings, indicating that the effects of any potential contamination are minimal.
>
> **4. Soften writing where needed**
> We thank the reviwer for this feedback. We have gone through the manuscript and softened statements that were too strong. Some examples:
>
> > *(Abstract)*
> > **Old:** our work provides the **empirical foundation** necessary to efficiently deploy robust code verifiers, thereby enabling their wider adoption
> > **New:** our work **offers empirical guidance toward** training cost-efficient code verifiers and **takes a step toward** their wider adoption
> >
> > *(Introduction, Contribution bullet 2)*
> > **Old:** We **isolate** the contributions of three core RLVR components
> > **New:** We **ablate** the contributions of three core RLVR components
> >
> > *(Section 4, Para 1)*
> > **Old:** This design choice **eliminates** confounding factors, **ensuring** observed performance reflects each component's individual contribution
> > **New:** This design choice **minimizes** confounding factors **to isolate each component's individual contribution as much as possible**
> >
> > *(Conclusion)*
> > **Old:** [...] DPO-Think-14B is an **optimal choice** for training verifiers for all scenarios.
> > **New:** [...] DPO-Think-14B is an **attractive choice** when a **substantial decrease in cost is worth more than a small drop in performance.**
> >
> > **Old:** our work establishes a **compute-optimal roadmap** for practitioners.
> > **New:** our work establishes a **practical cost-performance guide** for practitioners.
> >
> > *(Overall)* Replaced "contamination-free" with "decontaminated", paired with a limitation acknowledging potential contamination from earlier training stages.

---

> > ### Comment · Reviewer_yrvP · 2026-07-15
> > **Softened claims, confidence interval reporting and new section on downstream evaluation look good, recommend accept.**
> >
> > Thank you so much for your response.
> > I was already positive about the paper, and was mainly concerned with the stronger claims made.
> > I also want to thank the other reviewers for catching some of these claims that I missed. I especially appreciate the inclusion of the statistical reporting of the confidence intervals and the new section on downstream evaluation that came about because of that dialog between the other reviewers and the authors.
> >
> > Of course, it would be great to have experiments with bigger models, but I understand that this is difficult with compute costs, and so I personally think if the writing acknowledges the limitation properly (and the representative changes posted by the authors seem to do so, maybe a couple of lines more acknowledging this again can't hurt), this would not be a reason for me to not recommend the paper to be accepted.
> >
> > Overall I think that with the softened claims, the inclusion of a clearer limitations section, more statistical reporting, and the new section on downstream integration, I am happy with the state of the paper and would like to thank the authors again for engaging in a positive dialogue and making these changes.

---

### Review · Reviewer_nHNn · 2026-07-05

**Summary Of Contributions:**

This paper presents a controlled ablation study of three components that distinguish full RLVR training from cheaper alternatives when training code verifiers: (1) intermediate thinking traces, (2) learning from negative samples, and (3) on-policy training. The paper introduces Aletheia, a contamination-free testbed built from CodeContests+ with execution-grounded pass rates, structured into four evaluation splits (Heldout, Strong, Hard, Adv). Verifiers are evaluated on two metrics: ListAcc (BoN selection) and Kendall's $\tau$ (ranking reconstruction, as a proxy for RL reward quality), across 1.5B/7B/14B model scales.

**Audience:**

Yes

**Audience Explanation:**

Researchers working on reward modeling, RLVR, and code generation post-training would find this relevant.

**Claims And Evidence:**

Yes

**Claims Explanation:**

The paper addresses an important gap: RLVR recipes for verifiers/reward models are increasingly central to post-training pipelines, but their cost makes ablation prohibitive for most groups, and no such controlled study previously existed for the code domain specifically.

Aletheia is a well-conceived testbed: execution-grounded, listwise, and constructed to induce three practically relevant OOD shifts (generator capability, near-correct distractors, adversarial perturbations).

The paper links training-time design choices (thinking, negatives, on-policy) to two distinct downstream roles of verifiers (BoN selection vs RL reward ranking), and reports different behaviors across scales.

**Requested Changes:**

1. Lack of statistical analysis; it reports a single run with no variance across seeds. The paper should report results with mean and standard deviation across multiple runs.
2. All main results initialize from DeepSeek-R1-Distill-Qwen2.5 at 1.5/7/14B, that is, a single model family at three scales. Discuss/acknowledge the single model-family limitation for the core scale ablations, and ideally include at least one additional base model family (even at a single scale).

---

> ### Author Response · Authors · 2026-07-10
> **Response to reviewer nHNn (1/1)**
>
> We thank the reviewer for appreciating our work and highlighting its relevance to the community. We would also like to thank them for taking the time to provide relevant feedback, which we address as follows.
>
> **Lack of statistical analysis**
> We thank the reviewer for this suggestion. Unfortunately, retraining each of the 21 configurations across multiple seeds is computationally prohibitive, costing ~14,600 GPU hours per seed (estimated using the costs reported in Tables 3-5 for 781 steps). However, we can tractably quantify the stochasticity during evaluation. We report 95% confidence intervals for all tables over 16 generations per prompt at temperature=0.6 and top-p=0.95. This change is reflected in the revised manuscript.
>
> **Add another model family**
> We thank the reviewer for their feedback. We agree that a second family of models would make our conclusions more reliable. We chose the DeepSeek-R1-Distill-Qwen models because they are available at a wide range of model sizes and have already been warm-started to generate thinking traces. We are currently running the requested experiment using the **Olmo3-7B** family of models, which similarly provide separate *-Thinking* and *-Instruct* models. However, due to limited compute, these runs will take ~10 days to complete. We will report these results and integrate them into the manuscript once the runs are completed.